



Satellite-based radiative forcing by light-absorbing particles in snow across the
Northern Hemisphere
Jiecan Cui[1], Tenglong Shi[1], Yue Zhou[1], Dongyou Wu[1], Xin Wang[1,2] and Wei Pu[1]
[1]Key Laboratory for Semi-Arid Climate Change of the Ministry of Education, College
of Atmospheric Sciences, Lanzhou University, Lanzhou 730000, China
[2]Institute of Surface-Earth System Science, Tianjin University, Tianjin 300072, China
Corresponding author: Wei Pu (puw09@lzu.edu.cn) and Xin Wang (wxin@lzu.edu.cn)





**Abstrac**t. Snow is the most reflective natural surface on Earth and consequently plays
an important role in Earth's climate. Light-absorbing particles (LAPs) deposited on the
snow surface can effectively decrease snow albedo, resulting in positive radiative
forcing. In this study, we used remote sensing data from NASA's Moderate Resolution
Imaging Spectroradiometer (MODIS) and the Snow, Ice, and Aerosol Radiative
(SNICAR) model to quantify the reduction in snow albedo due to LAPs, before
validating and correcting the data against *in situ* observations. We then incorporated
these corrected albedo-reduction data in the Santa Barbara DISORT Atmospheric
Radiative Transfer (SBDART) model to estimate Northern Hemisphere radiative
forcing in January and February for the period 2003–2018. Our analysis reveals an
average corrected reduction in snow albedo ($\Delta\alpha_{MODIS,corrected}^{LAPs}$) of ~0.0246, with
instantaneous radiative forcing ($RF_{MODIS,ins}^{LAPs}$) and daily radiative forcing ($RF_{MODIS,daily}^{LAPs}$)
values of ~5.9 and ~1.7 W m$^{-2}$, respectively. We also observed significant spatial
variations in $\Delta\alpha_{MODIS,corrected}^{LAPs}$, $RF_{MODIS,ins}^{LAPs}$, and $RF_{MODIS,daily}^{LAPs}$ throughout the
Northern Hemisphere, with the lowest respective values (~0.0123, ~1.1 W m$^{-2}$, and
~0.29 W m$^{-2}$) occurring in the Arctic and the highest (~0.1669, ~36 W m$^{-2}$, and ~11
W m$^{-2}$) in northeastern China. From MODIS retrievals, we determined that the LAP
content of snow accounts for 57.6% and 37.2% of the spatial variability in Northern
Hemisphere albedo reduction and radiative forcing, respectively. We also compared
retrieved radiative forcing values with those of earlier studies, including local-scale
observations, remote-sensing retrievals, and model-based estimates. Ultimately,



estimates of radiative forcing based on satellite-retrieved data are shown to represent
true conditions on both regional and global scales.
**1.   Introduction**
Seasonal snow cover affects 30% of Earth's land surface and exerts a cooling influence
on global climate through its direct interaction with the surface radiation budget
(Painter et al., 1998; Flanner et al., 2011). However, snow surface darkening due to
light-absorbing particles (LAPs) such as black carbon (BC), organic carbon (OC), dust,
and algae, can significantly alter the reflective properties of snow (Warren, 1982, 1984;
Hadley and Kirchstetter, 2012). When deposited on the snow surface, LAPs increase
the absorption of solar radiation (Painter et al., 2012a; Liou et al., 2014; Dang, 2017),
thereby reducing the snow albedo (Warren and Brandt, 2008; Kaspari et al., 2014). As
a result, radiative forcing of LAPs in snow (RFLS) plays a critical role in snow-cover
decline on both regional and global scales (Warren and Wiscombe, 1980), perturbing
the climate system and impacting hydrological cycles (Qian et al., 2011).
One of the primary LAPs, BC, is derived from the incomplete combustion of fossil
fuels and biomass (Bond et al., 2013; Dang et al., 2015) and is second only to $CO_2$ in
its contribution to climate forcing (Hansen and Nazarenko, 2004; Ramanathan and
Carmichael, 2008; Bond et al., 2013). Yet, despite considerable efforts to measure the
BC content of Northern Hemisphere snow and ice (Doherty et al., 2010, 2014; Huang
et al., 2011; Ye et al., 2012; Wang et al., 2013b, 2017), the inherent challenges presented



by a temporospatially variable snow cover mean our understanding of LAPs in snow is
far from complete. As a result, RFLS estimate based on field measurements remains a
persistent uncertainty in regional and global scale (Zhao et al., 2014).
Several previous investigations have utilized numerical models to estimate RFLS,
including that of Hansen and Nazarenko (2004), who concluded that BC in snow and
ice exerts a positive climate forcing throughout the Northern Hemisphere of +0.3
W m$^{-2}$, or approximately one quarter of observed global warming. More recently,
Flanner et al. (2007) employed an aerosol/chemical-transport general-circulation model,
coupled with the Snow, Ice, and Aerosol Radiative (SNICAR) model (Flanner et al.,
2007; 2009), to estimate globally averaged radiative forcing values of +0.054 (range
0.007–0.13) and +0.049 (0.007–0.12) W m$^{-2}$ for a strong (1998) and weak (2001) boreal
fire year, respectively. Using the Weather Research and Forecasting (WRF) model
(Skamarock et al., 2008) coupled with a chemistry component (Chem) (Grell et al.,
2005) and SNICAR modeling, Zhao et al. (2014) demonstrated that RFLS over northern
China in January–February 2010 was ~10 W m$^{-2}$. However, despite their potentially
valuable contribution, climate models contain significant uncertainties in
representations of LAP emissions, transport, deposition, and post-depositional
processes that can propagate into simulations of LAP concentrations and their climate
forcing (Qian et al., 2015; Lee et al., 2016). Zhao et al. (2014) also confirmed that,
relative to observational data, modeled LAPs and radiative forcing estimates exhibit
biases that are difficult to explain and quantify. These shortcomings underscore the need



for a refined approach to estimating real-time RFLS that minimizes the mismatch
between field observations and model simulations.
In addition to modeling, remote sensing has been used to assess the physical
characteristics of snow cover (Nolin and Dozier, 1993, 2000; Painter et al., 2009, 2012a,
2013; Miller et al., 2016). Nolin and Dozier (2000), for example, retrieved grain-size
data from satellite-derived reflectance at near-infrared (NIR) wavelengths, following
the rationale that snow-grain size, in conjunction with solar zenith angle, dictates the
path-length of penetrating photons (Wiscombe and Warren, 1980) and thus influences
albedo in the NIR. Similarly, recent studies have attempted to employ satellite-derived
snow albedo at visible (VIS) wavelengths to retrieve RFLS data (Seidel et al., 2016; Pu
et al., 2019). Briefly, this retrieval method exploits the imaginary component of the
complex refractive index for ice ($K_{ice}$), which is very low at VIS wavelengths and
results in the extremely high VIS albedo for pure snow. In contrast, the imaginary
component of the complex refractive index for LAPs ($K_{LAPs}$) at VIS wavelengths is
orders of magnitude greater, resulting in the reduction in VIS snow albedo (Wiscombe
and Warren, 1980). Moreover, albedo variability at VIS wavelengths is dominated by
even minor concentrations of LAPs (Brandt et al., 2011; Painter et al., 2012b).
Painter et al. (2012a) employed surface-reflectance data provided by NASA's Moderate
Resolution Imaging Spectroradiometer (MODIS) for the Upper Colorado River Basin
and Hindu Kush-Himalaya (HKH) to make the first quantitative, remote-sensing-based
retrievals of instantaneous surface radiative forcing (RF) due to LAPs. Relative to the



Western Energy Balance of Snow (WEBS) network (Painter et al., 2007), that study
established that MODIS-derived radiative forcing exhibits a positive bias at lower RF
values and a slightly negative bias at higher values. A more recent study by Seidel et al.
(2016) used remote sensing to constrain instantaneous melt-season RFLS values of 20–
200 W m$^{-2}$ for the Sierra Nevada and Rocky Mountains, while Pu et al. (2019) reported
MODIS-derived values of 22–65 W m$^{-2}$ for northern China in January–February
(regional average ~45 W m$^{-2}$). Acknowledging this demonstrated efficacy of remote
sensing retrievals for establishing RFLS on regional scales, we note this approach has
so far not captured spatial variability in RFLS on a global scale.
In this study, we employed MODIS data to determine the reduction in Northern
Hemisphere snow albedo due to LAPs. Retrievals were validated and corrected
according to ground-based snow observations, after which spatial variability in albedo
reduction and radiative forcing were assessed quantitatively. Finally, we compared our
satellite-derived radiative forcing values with the results of a Global Climate Model
(GCM) (Flanner et al., 2009) and the Coupled Model Intercomparison Project Phase 6
(CMIP6) (Eyring et al., 2016). Despite the persistence of non-negligible uncertainties
and biases, our satellite-based retrievals constitute the first hemisphere-scale
assessment of RFLS and provide valuable parameters for improving climate model
simulations.
**2.   Data**



## 2.1. Remote-sensing data

To investigate the impact of LAPs on snow albedo, we utilized the following MODIS data sets: surface albedo (MCD43C3; 0.05° × 0.05° resolution), snow cover (MYD10C1; 0.05° × 0.05° resolution), land cover type (MCD12C1; 0.05° × 0.05° resolution), and atmospheric parameters (MYD08_D3; 1° × 1° resolution). Each data set corresponds to January–February for the period 2003–2018 (https://earthdata.nasa.gov, last access: 20 January 2019). MCD43C3 is the daily combined MODIS output derived from both the Terra and Aqua satellites, and provides black-sky albedo (directional hemispherical reflectance, DHF) and white-sky albedo (bi-hemispherical reflectance, BHF) at local solar noon for bands 1–7 (band 1, 620–670 nm; band 2, 841–876 nm; band 3, 459–479 nm; band 4, 545–565 nm; band 5, 1230–1250 nm; band 6, 1628–1652 nm; band 7, 2105–2155 nm), as well as values for quality control, local noon solar zenith angle, and associated parameters. MCD43C3 observations are weighted to estimate albedo on the 9th day of each 16-day period and have been corrected for the influence of local slope and aspect, atmospheric gases, and aerosols.

Snow-cover data are provided daily by MYD10C1 as a report of the snow-cover fraction (SCF), generated by the Normalized Difference Snow Index (NDSI). MCD12C1 provides a spatially aggregated and reprojected land-cover type, which is derived from the supervised classification of MODIS reflectance data, while MODIS MYD08_D3 reports values of solar azimuth angle.



Average-daily solar radiation was obtained from NASA's Clouds and the Earth's
Radiant Energy System (CERES: https://ceres.larc.nasa.gov, last access: 12 April 2019),
part of the Earth Observing System comprising the Aqua, Terra, and S-NPP satellites.
CERES provides instantaneous measurements of solar radiation, which are then
converted to average-daily flux by angular dependence and empirical diurnal albedo
modeling as the satellite passes through the point of descent (Doelling et al., 2013; Su
et al., 2015; Loeb et al., 2018). We used the downward shortwave flux at the surface,
provided by the "CERES Single Scanner Footprint 1.08 (SSF1deg)" product, to
estimate average-daily RFLS under clear-sky conditions.
Shuttle Radar Topography Mission (SRTM) digital elevation data are provided by the
US Geological Survey (https://www.usgs.gov/, last access: 9 December 2018) to adjust
slope- and aspect-induced changes of surface solar radiation in complex terrain. The
spatial resolution of SRTM data for the Northern Hemisphere is 30 m.
**2.2. Snow depth data**
Estimates of snow depth were obtained from the European Centre for Medium-Range
Weather     Forecasts     (ECMWF)     Interim     Re-Analysis     (ERA-Interim)
(https://www.ecmwf.int, last access: 15 January 2019). ERA-Interim is a new
generation of reanalysis based on a 12-hourly and 4-dimensional variational data
assimilation (4D-Var) covering the period 1979–present. ERA-Interim performs better
in model physics frameworks, data quality control, and background error criteria than



previous versions (Berrisford et al., 2011; Brun et al., 2013). In this study, we used
snow-water equivalent (SWE) data for January–February covering the period 2003–
2018. These data were generated by forecast models and updated according to a
Cressman analysis of snow observations (Drusch et al., 2004; Dee et al., 2011). We note
that the previous occurrence of false snow-free patches, arising from application of
Cressman analysis in regions of sparse ground control, has been mitigated by ECMWF
upgrades (Dee et al., 2011). Finally, SWE is converted to snow depth by assuming that
average January–February snow density is ~200 kg m$^{-3}$, consistent with snow-depth
estimates by the Canadian Meteorological Centre (CMC) (Sturm et al., 1995; Brown
and Mote, 2009).
**2.3. BC emission and deposition data**
We used the PKU-BC-2007 (EPKU: http://inventory.pku.edu.cn, last access: 5 June
2019) global inventory, with a spatial resolution of 0.1° × 0.1°, to obtain BC emissions
data for the period 2003–2014. PKU-BC-2007 was developed using a bottom-up
method based on sub-national fuel-combustion data (Wang et al., 2013a, 2014b) and an
updated set of BC emission factors (Wang et al., 2012). To constrain BC deposition
fluxes during our study period, we applied the Modern-Era Retrospective Analysis for
Research        and        Applications,        version        2        (MERRA-2:
https://gmao.gsfc.nasa.gov/reanalysis/MERRA-2/, last access: 5 June 2019), which
simulates BC via a radiatively coupled version of the Goddard Chemistry, Aerosol,



Radiation, and Transport model (GOCART; Chin et al. 2002; Colarco et al. 2010). The
efficacy of the GOCART aerosol module in simulating observable aerosol
characteristics has been verified by a wealth of previous studies (e.g., Nowottnick et al.
2010, 2011; Bian et al. 2013; Randles et al., 2017).

## 6    2.4. Climate model simulations

We compared our remotely sensed retrievals of snow BC for the 2003–2014 study
period with simulated concentrations derived from CMIP6 (https://esgf-node.llnl.gov/,
last access: 15 July 2019), which coordinates the design and distribution of GCM
simulations of past, present, and future climate. To date, only two CMIP participants,
CESM2 and CESM2-WACCM, have provided simulations of snow BC concentrations.
Therefore, we employed data derived from the CESM2 and CESM2-WACCM
historical experiments, in conjunction with ERA-Interim SWE, MODIS-retrieved snow
grain-size, and CERES downward shortwave flux data, to model RFLS for the study
period. Simulations were performed using the Snow, Ice, and Aerosol Radiative
(SNICAR) and Santa Barbara DISORT Atmospheric Radiative Transfer (SBDART)
models, and modeled output compared to satellite-based retrievals. We also compared
our retrieval-based estimates of RFLS to values simulated by the SNICAR model
coupled with a GCM (Flanner et al., 2007, 2009).

## 21   3.    Methods



## 3.1. Radiative transfer model

In this study, we used the Santa Barbara DISORT Atmospheric Radiative Transfer (SBDART) model to calculate surface solar irradiance. Constituting one of the most widely applied models for calculating the atmospheric radiative transfer at Earth's surface, under both clear- and cloudy-sky conditions (Ricchiazzi et al., 1998), SBDART combines a low-resolution atmospheric transmission model, Discrete Ordinate Radiative Transfer (DISORT) module, and Mie scattering output for the scattering of light by ice crystals and water droplets (Stamnes et al., 1988; Fu et al., 2017). Radiative transfer equations for a vertically inhomogeneous, non-isothermal, plane-parallel atmosphere are integrated numerically using the DISORT module. SBDART comprises multiple standard atmospheric profiles, cloud models, basic surface types, standard types, as well as vertical distribution models for aerosols and gas absorption, and enables users to specify these input parameters in real values. In our study, we used SBDART to calculate the clear-sky spectral direct and diffuse solar radiation at local noon. Spectral radiation ranges from 0.3 to 1.3 μm, at 0.01 μm intervals, and with 1° latitude resolution. Average incident direct and diffuse solar spectra for January–February are shown in Fig. S1.

The Snow, Ice, and Aerosol Radiative (SNICAR) model is a two-stream multiple scattering radiative transfer model (Flanner et al., 2007, 2009) that has been used widely both to simulate the albedo, transmission, and vertical absorptivity of LAP-contaminated snowpack and to estimate RFLS (Painter et al., 2012a; Bryan et al., 2013;





Miller et al., 2016). SNICAR employs the theory proposed by Wiscombe and Warren
(1980) and Toon et al. (1989). Specifically, snow is considered to be composed of
aggregated ice spheres with optical effective radii ($R_{eff}$) of 50–1500 μm and lognormal
distribution. SNICAR also accounts for incident radiation, surface spectral distribution,
solar zenith angle, snow depth and density, snow layer number, and the type and
concentration of LAPs in the snowpack. The model's ability to provide realistic
simulations of snow albedo has been verified by several previous studies (Hadley and
Kirchstetter, 2012; Meinander et al., 2013; Zhong et al., 2017; Wang et al., 2017).
**3.2. Retrieval of quantitative snow properties from remote sensing**
The variability of spectral snow albedo depends on the LAP content, grain size, and
depth of the snowpack, in addition to solar zenith angle. As shown in Fig. 1a, the
deposition of BC (as representative of LAPs generally) serves to decrease the albedo of
snow significantly, particularly in the ultraviolet (UV) and VIS wavelengths, which
account for approximately half of all direct solar irradiation and the majority of diffuse
solar irradiation (Fig. S1). In contrast, the impact of BC on albedo is considerably
smaller in NIR wavelengths and can be negligible at >~1150 nm. Snow depth plays a
similar role to LAP content and primarily affects albedo in UV and VIS wavelengths
(Fig. 1c).
Although snow albedo decreases with snow depth, previous studies have tended to
assume a semi-infinite snowpack for which albedo is independent of depth. As a



consequence, the role of LAPs in albedo reduction has been overestimated for those
areas where the snowpack is thin (Warren, 2013). In this study, we incorporated ERA-
Interim SWE data in our SNICAR model simulations to correct for the snow-depth
overestimation effect. In contrast, snow grain-size and solar zenith angle influence the
snow albedo chiefly in NIR wavelengths (Fig. 1b, d). Specifically, albedo tends to
decrease with increasing snow grain-size and declining solar zenith angle. In this study,
we derived quantitative snow parameters (grain size, albedo reduction, and RFLS) from
MODIS data in conjunction with the SNICAR and SBDART models. The specific
workflow for retrieving RFLS from satellite data is shown in Fig. 2.
**3.2.1.   Retrieval of blue-sky albedo**
The actual spectral albedo for a land surface at wavelength $\lambda$ (also called blue-sky
albedo: $\alpha_{blue-sky,\lambda}$) can be calculated as follows:
$$\alpha_{blue-sky,\lambda} = f_{dif,\lambda} \cdot \alpha_{white-sky,\lambda} + (1 - f_{dif,\lambda}) \cdot \alpha_{black-sky,\lambda} \qquad (1)$$
where $\alpha_{white-sky,\lambda}$ and $\alpha_{black-sky,\lambda}$ are MODIS-derived values for white-sky and
black-sky albedo, respectively, and $f_{dif,\lambda}$ is the ratio of diffuse radiation to the total
solar radiation flux (Lewis and Barnsley, 1994). The latter is calculated as follows:
$$f_{dif,\lambda} = \frac{E_{dif}(\lambda; \varphi)}{E_{dif}(\lambda; \varphi) + E_{dir}(\lambda; \varphi)} \qquad (2)$$
where $\varphi$ is latitude, and $E_{dif}(\lambda; \varphi)$ and $E_{dir}(\lambda; \varphi)$ denote diffuse and direct
spectral solar radiation, respectively, derived from the SBDART model.



**3.2.2. Retrieval of snow cover and albedo values**
As shown in Fig. 2, the snow-covered area is mapped according to the blue-sky albedo
($\alpha_{blue-sky}$) in band 4 (band center ~555 nm: $\alpha_{blue-sky,\lambda}(\lambda_{VIS})$) and the Normalized
Difference Snow Index (NDSI), both of which exceed 0.6 (Negi and Kokhanovsky,
2011). NDSI is calculated as follows (Dozier and Marks, 1987; Hall et al., 1995):

$$\text{NDSI} = \frac{\alpha_{blue-sky,\lambda}(\lambda_{VIS}) - \alpha_{blue-sky,\lambda}(\lambda_{SWIR})}{\alpha_{blue-sky,\lambda}(\lambda_{VIS}) + \alpha_{blue-sky,\lambda}(\lambda_{SWIR})} \quad (3)$$

where $\alpha_{blue-sky,\lambda}(\lambda_{SWIR})$ is $\alpha_{blue-sky,\lambda}$ in band 6 (band center ~1640 nm).
According to the MODIS Snow Products Collection 6 User Guide
(http://nsidc.org/data), the Fractional Snow Cover ($FSC$) can be calculated as follows:

$$FSC = -0.01 + 1.45 \cdot \text{NDSI} \quad (4)$$

Accordingly, the identified snow-covered area (ISCA) has an $FSC$ value of >86% but
not always 100%. Therefore, the MODIS-derived albedo for a particular ISCA is a
combination of values representing both snow and the snow-free underlying surface.
Following Pu et al. (2019), the snow albedo ($\alpha_{snow,\lambda}^{MODIS}$) can be distinguished from the
mixed albedo by the equation:

$$\alpha_{blue-sky,\lambda} = \frac{E_\lambda \cdot FSC \cdot \alpha_{snow,\lambda}^{MODIS} + E_\lambda \cdot (1 - FSC) \cdot \alpha_{underlying,\lambda}}{E_\lambda}$$

$$= FSC \cdot \alpha_{snow,\lambda}^{MODIS} + (1 - FSC) \cdot \alpha_{underlying,\lambda} \quad (5)$$

$$\alpha_{snow,\lambda}^{MODIS} = \frac{\alpha_{blue-sky,\lambda} - (1 - FSC) \cdot \alpha_{underlying,\lambda}}{FSC} \quad (6)$$





where $E_\lambda$ is total solar radiation. $\alpha_{underlying,\lambda}$ represents the albedo of the
underlying surface and was obtained from Siegmund and Menz (2005). As depicted in
Fig. 3b, vegetation and bare soil are the main types of underlying surface in the ISCA.

### 3.2.3. Retrieval of snow grain size

The snow optical-equivalent grain size ($R_{eff}$) is retrieved by fitting SNICAR-simulated
snow albedo to MODIS-derived snow albedo at 1240 nm (the central wavelength of
MODIS band 5), following the protocol of Nolin and Dozier (2000). This retrieval
method is not influenced by liquid water and water vapor and has been employed
widely in previous studies (e.g., Painter et al., 2013; Seidel et al, 2016). Both Nolin and
Dozier (2000) and Pu et al. (2019) reported that the retrieved $R_{eff}$ compares favorably
with ground-based measurements of snow grain size. In this study, we chose to exclude
the ISCA, where MODIS-derived snow albedo at 1240 nm is <0.3, to avoid
misrepresenting $R_{eff}$ (Tedesco et al., 2007).

### 3.2.4. Retrieval of snow albedo reduction

The instantaneous, spectrally integrated reduction in snow albedo due to LAPs
($\Delta\alpha_{MODIS,ins}^{LAPs}$) is estimated for local-noon and clear-sky conditions, using solar radiation
and the difference between MODIS-derived spectral snow albedo ($\alpha_{snow,\lambda}^{MODIS}$) and
simulated pure snow albedo ($\alpha_{snow,\lambda}^{mdl}$). Because MODIS provides only four VIS bands,
we fitted snow albedo data obtained via MODIS to a continuous 300–1300 nm spectrum





(with a 10 nm interval) following the method provided by Pu et al. (2019). Thereafter,
$\Delta\alpha_{MODIS,ins}^{LAPs}$ can be calculated as follows:
$$\Delta\alpha_{MODIS,ins}^{LAPs} = \frac{\sum_{\lambda=300nm}^{\lambda=1300nm}\left(\alpha_{snow,\lambda}^{mdl} - \alpha_{snow,\lambda}^{MODIS}\right)\cdot(E_{dir,\lambda}\cdot\cos\beta + E_{dif,\lambda})\cdot\Delta\lambda}{\sum_{\lambda=300nm}^{\lambda=1300nm}(E_{dir,\lambda}\cdot\cos\beta + E_{dif,\lambda})\cdot\Delta\lambda} \qquad (7)$$
where $\alpha_{snow,\lambda}^{mdl}$ is the pure snow albedo simulated by SNICAR using MODIS-derived
$R_{eff}$ and ERA-Interim snow depth data, $\alpha_{snow,\lambda}^{MODIS}$ is the continuous snow albedo
derived from MODIS retrievals, and $\Delta\lambda$ is 10 nm. Finally, $\beta$ represents local solar
zenith angle, which is obtained using the topographic correction method (Teillet et al.,
1982; Negi and Kokhanovsky, 2011):
$$\cos\beta = \cos\theta_0\cos\theta_T + \sin\theta_0\sin\theta_T\cos(\phi_0 - \phi_T) \qquad (8)$$
for which $\theta_0$ represents the solar zenith angle for a horizontal surface, $\phi_0$ is the solar
azimuth angle, and $\theta_T$ and $\phi_T$ denote slope inclination and aspect, respectively.
Similarly, we used measurements of LAPs in contaminated snow to calculate the
instantaneous, *in situ* reduction in snow albedo ($\Delta\alpha_{in-situ,ins}^{LAPs}$). To derive a correction
factor for MODIS retrievals, we applied a similar validation strategy to that of Zhu et
al. (2017):
$$c = \frac{1}{n}\sum_{i=1}^{n}\left(\frac{\Delta\alpha_{MODIS,ins}^{LAPs}}{\Delta\alpha_{in-situ,ins}^{LAPs}}\right) \qquad (9)$$
where c is the correction factor for $\Delta\alpha_{MODIS,ins}^{LAPs}$ and n is the number of the
respective *in situ* measurements. Accordingly, the corrected albedo reduction
($\Delta\alpha_{MODIS,corrected}^{LAPs}$) is calculated as follows:
$$\Delta\alpha_{MODIS,corrected}^{LAPs} = \frac{1}{c}\cdot\Delta\alpha_{MODIS,ins}^{LAPs} \qquad (10)$$



**3.2.5. Retrieval of RFLS**
The instantaneous, spectrally integrated RFLS ($RF_{MODIS,ins}^{LAPs}$) is calculated for noon and
clear-sky conditions as follows:
$$RF_{MODIS,ins}^{LAPs} = \Delta\alpha_{MODIS,corrected}^{LAPs} \cdot \sum_{\lambda=300\,nm}^{\lambda=1300\,nm} \cdot (E_{dir,\lambda} \cdot \cos\beta + E_{dif,\lambda}) \cdot \Delta\lambda$$

6 (11)

We assumed that the properties for snow and LAPs remain invariable throughout the
day and that the average-daily RFLS ($RF_{MODIS,daily}^{LAPs}$) can be expressed as follows:
$$RF_{MODIS,daily}^{LAPs} = \Delta\alpha_{MODIS,corrected}^{LAPs} \cdot (SW_{dir} \cdot \cos\beta + SW_{dif}) \quad (12)$$
where $SW_{dir}$ and $SW_{dif}$ represent the average-daily direct and diffuse downward
shortwave fluxes, respectively, obtained from CERES under clear-sky conditions.
**3.2.6. Attribution of spatial variability in snow albedo reductions and radiative**
**forcing**
As demonstrated above, reductions in snow albedo and RFLS are dependent primarily
on LAP content, $R_{eff}$, snow depth ($SD$), solar zenith angle, surface topography, and
solar radiation, the latter three of which can be categorized as the geographic factor ($G$).
We used an impurity index ($I_{LAPs}$) to represent the LAP content of the snowpack (Di
Mauro et al., 2015; Pu et al., 2019), following the equation:
$$I_{LAPs} = \frac{\ln(\alpha_{snow,band4}^{MODIS})}{\ln(\alpha_{snow,band5}^{MODIS})} \quad (13)$$



where $\alpha_{snow,band4}^{MODIS}$ and $\alpha_{snow,band5}^{MODIS}$ are the MODIS-derived snow albedo values for
bands 4 and 5, respectively. We then calculated $\Delta\alpha_{MODIS,corrected}^{LAPs}$ as follows:
$$\Delta\alpha_{MODIS,corrected}^{LAPs} = f(I_{LAPs}, R_{eff}, SD, G) \tag{14}$$
Values for $R_{eff}$, $SD$, and $G$ were kept spatially constant as the averages $\overline{R_{eff}}$, $\overline{SD}$,
and $\bar{G}$, with $\bar{G}$ requiring spatially constant values for the solar zenith angle, surface
topography, and solar radiation parameters. As a result, spatial variability in snow
albedo reduction due to $I_{LAPs}$ can be expressed as
$$\Delta\alpha_{MODIS,corrected}^{LAPs}(I_{LAPs}) = f(I_{LAPs}, \overline{R_{eff}}, \overline{SD}, \bar{G}) \tag{15}$$
The following three equations were applied in a similar manner:
$$\Delta\alpha_{MODIS,corrected}^{LAPs}(R_{eff}) = f(\overline{I_{LAPs}}, R_{eff}, \overline{SD}, \bar{G}) \tag{16}$$
$$\Delta\alpha_{MODIS,corrected}^{LAPs}(SD) = f(\overline{I_{LAPs}}, \overline{R_{eff}}, SD, \bar{G}) \tag{17}$$
$$\Delta\alpha_{MODIS,corrected}^{LAPs}(G) = f(\overline{I_{LAPs}}, \overline{R_{eff}}, \overline{SD}, G) \tag{18}$$
We then fitted $\Delta\alpha_{MODIS,corrected}^{LAPs}$ through multiple linear regression:
$$\Delta\alpha_{MODIS}^{LAPs,fit} = a \cdot \Delta\alpha_{MODIS,corrected}^{LAPs}(I_{LAPs}) + b \cdot$$
$$\Delta\alpha_{MODIS,corrected}^{LAPs}(R_{eff}) + c \cdot \Delta\alpha_{MODIS,corrected}^{LAPs}(SD) + d \cdot \Delta\alpha_{MODIS,corrected}^{LAPs}(G)$$

16   (19)

where $\Delta\alpha_{MODIS}^{LAPs,fit}$ is the fitted snow albedo reduction and a, b, c, and d denote the
regression coefficients. Figure S3a illustrates how $\Delta\alpha_{MODIS}^{LAPs,fit}$ can explain 98% of the
variance in $\Delta\alpha_{MODIS,corrected}^{LAPs}$. Therefore, the attribution of spatial variance in





$\Delta\alpha_{MODIS,corrected}^{LAPs}$ can be replaced with $\Delta\alpha_{MODIS}^{LAPs,fit}$, enabling Eq. (19) to be written as
follows:
$\Delta\alpha_{MODIS}^{LAPs,fit} - \overline{\Delta\alpha_{MODIS}^{LAPs,fit}} = a \cdot \left(\Delta\alpha_{MODIS,corrected}^{LAPs}(I_{LAPs}) - \right.$
$\overline{\Delta\alpha_{MODIS,corrected}^{LAPs}(I_{LAPs})}\Big) + b \cdot \left(\Delta\alpha_{MODIS,corrected}^{LAPs}(R_{eff}) - \right.$
$\overline{\Delta\alpha_{MODIS,corrected}^{LAPs}(R_{eff})}\Big) + c \cdot \left(\Delta\alpha_{MODIS,corrected}^{LAPs}(SD) - \right.$
$\overline{\Delta\alpha_{MODIS,corrected}^{LAPs}(SD)}\Big) + d \cdot \left(\Delta\alpha_{MODIS,corrected}^{LAPs}(G) - \overline{\Delta\alpha_{MODIS,corrected}^{LAPs}(G)}\right)$  (20)
and
$\Delta\alpha_{MODIS,anomaly}^{LAPs,fit} = a \cdot \Delta\alpha_{MODIS,corrected,anomaly}^{LAPs}(I_{LAPs}) + b \cdot$
$\Delta\alpha_{MODIS,corrected,anomaly}^{LAPs}(R_{eff}) + c \cdot \Delta\alpha_{MODIS,corrected,anomaly}^{LAPs}(SD) + d \cdot$
$\Delta\alpha_{MODIS,corrected,anomaly}^{LAPs}(G).$    (21)
According to Huang and Yi (1991) and Pu et al. (2019), the fractional contribution of
LAP content to the variability in snow albedo reduction ($R_{\Delta\alpha}^{LAPs}$) can be calculated as:
$\qquad R_{\Delta\alpha}^{LAPs} = \frac{1}{m}\sum_{j=1}^{m} \frac{\left(a \cdot \Delta\alpha_{MODIS,corrected,anomaly}^{LAPs}(I_{LAPs})_j\right)^2}{K_j}$    (22)
$\qquad K_j = \left(a \cdot \Delta\alpha_{MODIS,corrected,anomaly}^{LAPs}(I_{LAPs})_j\right)^2 + \Big(b \cdot$
$\Delta\alpha_{MODIS,corrected,anomaly}^{LAPs}(R_{eff})_j\Big)^2 + \left(c \cdot \Delta\alpha_{MODIS,corrected,anomaly}^{LAPs}(SD)_j\right)^2 +$
$\left(d \cdot \Delta\alpha_{MODIS,corrected,anomaly}^{LAPs}(G)_j\right)^2$    (23)
where $m$ denotes the length of the data set. Values for $R_{\Delta\alpha}^{Reff}$, $R_{\Delta\alpha}^{SD}$, and $R_{\Delta\alpha}^{G}$ can be
derived in the same way. Similarly, we can obtain the fractional contribution for
radiative forcing ($R_{RF}^{LAPs}$, $R_{RF}^{Reff}$, $R_{RF}^{SD}$, and $R_{RF}^{G}$).



**4. Results**
**4.1. Study area**
Figure 3a depicts the ISCA employed in this study. Most are located between 40°N and
55°N in Eurasia and North America, and are dominated by grassland and bare-soil
surfaces (Fig. 3b). Several mid–high-latitude regions that typically support a deep
snowpack, including Russia, western Europe, and eastern North America, are not
identified by MODIS as ISCA due to the broad distributions of forest and shrubland in
those areas (Fig. 3b). This pattern is supported by Bond et al. (2006), who demonstrated
that, under such vegetated conditions, LAPs in snow exert a relatively minor influence
on radiative forcing. In the Arctic, where the polar night renders satellite-mounted
sensors unable to detect radiation, only a small part of southern Greenland can be
identified as snow-covered during January and February.
As illustrated in Fig. 3a, ISCA can be separated into four general regions according to
geographical distribution and pollution conditions (Fig. 4a, b): northeastern China
(NEC), Eurasia (EUA), North America (NA), and the Arctic. The following analysis of
snow albedo reduction and RFLS concerns ISCA during the January–February study
period.
**4.2. Global characteristics**





Previous studies have highlighted the dominant role of BC in light absorption by snow
(Wang et al., 2013b; Dang et al., 2017). The spatial distribution of BC emissions density
for the Northern Hemisphere in January–February is shown in Fig. 4a. Emissions
density exhibits a strong spatial inhomogeneity, ranging from $<10^{-1}$ to $>10^4$ g km$^{-2}$
month$^{-1}$ over ISCA. The highest values occur in NEC, where the regional average of
10750 g km$^{-2}$ month$^{-1}$ is considerably higher than values for EUA (5643 g km$^{-2}$
month$^{-1}$) and NA (761 g km$^{-2}$ month$^{-1}$), and the lowest values occur in the Arctic
(average 76 g km$^{-2}$ month$^{-1}$). The wet and dry deposition of BC constitute the primary
mechanisms for BC accumulation in snow. As shown in Fig. 4b, the distribution of BC
deposition (i.e., the sum of dry and wet deposition) is similar to BC emissions density,
with the highest (3.26 $10^{-12}$ kg m$^{-2}$ s$^{-1}$) and lowest (1.21 $10^{-13}$ kg m$^{-2}$ s$^{-1}$) regional
averages corresponding to NEC and the Arctic, respectively. Both NA and EUA return
intermediate deposition values, with regional averages of 5.75 $10^{-13}$ kg m$^{-2}$ s$^{-1}$ and 1.78
$10^{-12}$ kg m$^{-2}$ s$^{-1}$, respectively. Together, these data indicate that the NEC snowpack is
heavily polluted, and thus RFLS is likely to be highest, while the Arctic snowpack is
the least contaminated.
In addition to LAP content, the physical properties of the snowpack, such as depth and
grain size, also impact snow albedo (Fig. 1). As depicted in Fig. 4c, the average
snowpack in NEC (0.19 m thick) is thinner than in both NA (0.26 m) and EUA (0.21
m), implying a greater impact of snow depth on snow albedo and radiative forcing in
NEC. The greatest snow depths occur in the Arctic (>1 m) and can be considered semi-



infinite, meaning that the impact of depth on albedo and radiative forcing is negligible.
Figure 4d shows the spatial distribution of MODIS-derived snow grain radius ($R_{eff}$).
In contrast to BC emissions density, BC deposition, and snow depth, $R_{eff}$ exhibits
minor spatial variability, with regional average values for NEC, EUA, NA, and the
Arctic of 235 μm, 227 μm, 252 μm, and 255 μm, respectably. These values align with
the findings of several previous studies (Painter et al., 2013; Seidel et al, 2016; Pu et
al., 2019) and imply that the contribution of $R_{eff}$ to spatial variability in snow albedo
reduction and radiative forcing is negligible.
According to Eq. (10) and (11), local solar radiation is an important factor for
determining RFLS. Figure 4e, f depicts the January–February averaged surface direct
and diffuse solar irradiance, respectively, under clear-sky conditions. Average direct
radiation values for EUA (138 W m$^{-2}$) and NA (147 W m$^{-2}$) are comparable to one
another but high relative to NEC (87 W m$^{-2}$), which lies at a generally higher latitude
(>40°). The lowest values occur in the Arctic (9 W m$^{-2}$) due to that region's extreme
latitude. For diffuse radiation, average fluxes for NEC, EUA, NA, and the Arctic are 32
W m$^{-2}$, 46 W m$^{-2}$, 36 W m$^{-2}$, and 4 W m$^{-2}$, respectively. In summary, these data indicate
a smaller radiative forcing in the Arctic than in the other three regions.
**4.3. Corrections based on *in situ* observations**
*In situ* observations of snow albedo reduction ($\Delta\alpha_{in-situ,ins}^{LAPs}$) were used to quantitatively
correct MODIS retrievals through comparison with MODIS-retrieved snow albedo



reduction ($\Delta\alpha_{MODIS,ins}^{LAPs}$). Figure S2 displays scatterplots of the ratios of $\Delta\alpha_{MODIS,ins}^{LAPs}$ to
$\Delta\alpha_{in-situ,ins}^{LAPs}$ ($r_{in-situ}^{MODIS}$) for each sampling sites (Ye et al., 2012; Wang et al., 2013b,
2017; Doherty et al., 2014). Briefly, for NA and EUA, where the snowpack is relatively
pure, values for $r_{in-situ}^{MODIS}$ mostly range between 1 and 12. In contrast, the heavily
polluted snowpack in NEC returns $r_{in-situ}^{MODIS}$ values ranging from 0.5 to 2.5, indicating
a negative correlation between the biases of $\Delta\alpha_{MODIS,ins}^{LAPs}$ and snow contamination, and
thus supporting the findings of previous studies (Painter et al., 2012a; Pu et al., 2019).
To improve the quality of MODIS retrievals, we developed the correction factors
$c_{clean}$ for the relatively pure snowpack conditions observed in the EUA, NA, and the
Arctic and $c_{polluted}$ for the impure conditions found in NEC. According to Eq. (9),
values for $c_{clean}$ and $c_{polluted}$ are 5.6 and 1.1, respectively. Hereafter, our analyses
are based on the corrected MODIS retrievals.
Figure 5 compares the corrected MODIS retrievals to measurement-based results. For
clean snow, the mean absolute error (MAE) of $\Delta\alpha_{MODIS,corrected}^{LAPs}$ relative to
$\Delta\alpha_{in-situ,ins}^{LAPs}$ is 0.0096, with a root mean square error (RMSE) of 0.0129,
corresponding to respective radiative forcing MAE and RMSE values of 3.0 W m$^{-2}$ and
3.8 W m$^{-2}$ for $RF_{MODIS,ins}^{LAPs}$ and 0.95 W m$^{-2}$ and 1.2 W m$^{-2}$ for $RF_{MODIS,daily}^{LAPs}$. For
polluted snow conditions, the MAE and RMSE are 0.0501 and 0.0622 for albedo
reduction, respectively, with respective radiative forcing MAE and RMSE values of 14
W m$^{-2}$ and 18 W m$^{-2}$ for $RF_{MODIS,ins}^{LAPs}$ and 4.4 W m$^{-2}$ and 5.5 W m$^{-2}$ for $RF_{MODIS,daily}^{LAPs}$.
Together, these results imply that the corrected MODIS retrievals are plausible.




Nevertheless, we note that the correction used in this study is spatially rough due to the
low density of *in situ* measurements, and that both the uncertainty and bias are non-
negligible. As a result, we conducted a comprehensive series of comparisons between
the MODIS-derived retrievals and values provided via surface measurements, model
simulations, and remote sensing (see Sect. 5). We concluded that further field-based
measurements of snow albedo are required to improve the quality of satellite retrievals.
**4.4. Spatial distributions of snow albedo reduction and radiative forcing**
Figure 6 shows the spatial distributions and statistics of MODIS-based
$\Delta\alpha_{MODIS,corrected}^{LAPs}$ , $RF_{MODIS,ins}^{LAPs}$ , and $RF_{MODIS,daily}^{LAPs}$ retrievals. On average,
$\Delta\alpha_{MODIS,corrected}^{LAPs}$, $RF_{MODIS,ins}^{LAPs}$ , and $RF_{MODIS,daily}^{LAPs}$ provide respective values of
0.0246, 5.9 W m$^{-2}$, and 1.7 W m$^{-2}$ for Northern Hemisphere ISCA. The highest
$\Delta\alpha_{MODIS,corrected}^{LAPs}$ occurs in NEC, where the regional average of ~0.1669 exceeds those
of EUA (~0.0210) and NA (~0.0181) by a factor of ~8–9. This feature reflects the
relatively high rate of winter-time emissions over NEC, which results in the highest
level of BC deposition over ISCA (Fig. 4a–b). In contrast, being located far from major
sources of pollution, the relatively clean Arctic snowpack returns the lowest
$\Delta\alpha_{MODIS,corrected}^{LAPs}$ (~0.0123) of the entire Northern Hemisphere.
Consistent with snow albedo reduction, the highest regional-average radiative forcing
occurs in NEC, with respective $RF_{MODIS,ins}^{LAPs}$ and $RF_{MODIS,daily}^{LAPs}$ values of ~36 W m$^{-2}$
and ~11 W m$^{-2}$, and the lowest regional average occurs in the Arctic, with $RF_{MODIS,ins}^{LAPs}$





and $RF_{MODIS,daily}^{LAPs}$ values of ~1.1 W m$^{-2}$ and ~0.29 W m$^{-2}$, respectively. As well as
receiving the lowest levels of pollution, the relatively low winter-time surface solar
radiation also contributes to the Arctic returning the smallest radiative forcing (Fig. 4e–
f). Regional-average radiative forcing for NA and EUA are both intermediate, with
$RF_{MODIS,ins}^{LAPs}$ ($RF_{MODIS,daily}^{LAPs}$) values of ~4.8 W m$^{-2}$ (~1.4 W m$^{-2}$) and ~5.3 W m$^{-2}$ (~1.5
W m$^{-2}$), respectively. Furthermore, because NA and EUA experience similar pollution
conditions and are located at similar latitudes, both regions exhibit comparable radiative
forcing.
On a regional level, NEC $\Delta\alpha_{MODIS,corrected}^{LAPs}$ falls primarily within the range ~0.1177–
0.2157, and intra-regional variability is relatively small due to pervasive heavy
pollution (Fig. 4). Compared to snow albedo reduction, the radiative forcing for NEC
exhibits a slightly greater spatial variability due to latitude-dependent differences in the
flux of surface solar radiation, with $RF_{MODIS,ins}^{LAPs}$ ($RF_{MODIS,daily}^{LAPs}$) ranging from ~24
W m$^{-2}$ to ~53 W m$^{-2}$ (~6.9 W m$^{-2}$ to ~16 W m$^{-2}$). In NA, where the principal ISCA are
located in southern Canada, the western US, and Central America Plains,
$\Delta\alpha_{MODIS,corrected}^{LAPs}$ tends to range between ~0.0071 and ~0.0309. The Central America
Plains exhibit the highest value of $\Delta\alpha_{MODIS,corrected}^{LAPs}$ (~0.045), with corresponding
$RF_{MODIS,ins}^{LAPs}$ and $RF_{MODIS,daily}^{LAPs}$ values of 14 W m$^{-2}$ and ~3.9 W m$^{-2}$, respectively. This
region also displays a clear gradient in $RF_{MODIS,ins}^{LAPs}$ ($RF_{MODIS,daily}^{LAPs}$), with values
increasing from <5 W m$^{-2}$ (<1 W m$^{-2}$) in the northwest to >15 W m$^{-2}$ (>4 W m$^{-2}$) in the
southeast, in line with previously reported observational data (Doherty et al., 2014).



In EUA, $\Delta\alpha_{MODIS,corrected}^{LAPs}$, $RF_{MODIS,ins}^{LAPs}$, and $RF_{MODIS,daily}^{LAPs}$ fall largely within the
respective ranges of ~0.0097–0.0352, ~2.4–9.9 W m$^{-2}$, and ~0.69–2.9 W m$^{-2}$. The
Middle East returns relatively high values for $\Delta\alpha_{MODIS,corrected}^{LAPs}$ (>0.04), $RF_{MODIS,ins}^{LAPs}$
(>12 W m$^{-2}$), and $RF_{MODIS,daily}^{LAPs}$ (>4 W m$^{-2}$), likely due to the combined effects of
elevated dust fluxes (Solomos et al., 2017) and high solar insolation at these low
latitudes. Similar to the Middle East, northwestern China also exhibits relatively high
values for $\Delta\alpha_{MODIS,corrected}^{LAPs}$ (>0.035), $RF_{MODIS,ins}^{LAPs}$ (>11 W m$^{-2}$), and $RF_{MODIS,daily}^{LAPs}$
(>3 W m$^{-2}$), while this pattern likely reflects the influence of anthropogenic BC in
addition to natural dust (Pu et al., 2017) (Fig. 4a–b). In contrast, Europe and Russia
return the relatively low $\Delta\alpha_{MODIS,corrected}^{LAPs}$ (<0.03), with a
$RF_{MODIS,ins}^{LAPs}$ ($RF_{MODIS,daily}^{LAPs}$) value of <10 W m$^{-2}$ (<3 W m$^{-2}$), reflecting the generally
low concentration of LAPs in this region. Finally, in the Arctic, where ISCA comprise
only a small part of southern Greenland (see Sect. 4.1), respective
$\Delta\alpha_{MODIS,corrected}^{LAPs}$, $RF_{MODIS,ins}^{LAPs}$, and $RF_{MODIS,daily}^{LAPs}$ values fall within the ranges
~0.0018–0.038, ~0.18–3.8 W m$^{-2}$, and ~0.046–0.98 W m$^{-2}$.
**4.5. Attribution to the spatial variability of snow albedo reduction and radiative**
**forcing**
Here, we address the attributions to the spatial variability of snow albedo reduction and
radiative forcing. As discussed in Sect. 3.2.6, the spatial variability in snow albedo
reduction and radiative forcing are largely dependent on LAP content, snow grain radius,





snow depth, and the geographic factor. Figure 7 illustrates the fractional contributions
of each factor within the study regions. For the Northern Hemisphere as a whole, LAPs
($I_{LAPs}$) is the greatest contributor (57.6%) to snow albedo reduction, followed by $SD$
(38.1%); $R_{eff}$ and $G$ have only a minor influence (2.7% and 1.6%, respectively) (Fig.
7a). This result confirms that the concentration of LAPs in the snowpack plays a
fundamental role in spatial variability of snow albedo reduction.
LAPs also constitute the dominant contributors to snow albedo reduction on a regional
scale, accounting for 76.2% of the Arctic signal and 54.4% in EUA, and are the second
largest contributor in both NEC (43.7%) and NA (48.1%). The contribution of $SD$ is
greatest in NEC (51.6%) and NA (49.8%), with slightly lower values in EUA (40.3%),
reflecting the significant spatial variability in $SD$ across these regions. In the Arctic,
the snowpack is sufficiently thick to be considered a homogeneous, semi-infinite
snowpack and thus the contribution of $SD$ is negligible. In contrast, $R_{eff}$ makes only
minor contributions in NEC (3.9%), NA (1.4%), and EUA (3.3%) but is an important
factor in the Arctic (22%). Finally, $G$ makes the smallest contribution to snow albedo
reduction (<2%), both on regional and global scales.
On a hemispheric scale, the greatest contributors to radiative forcing are LAP content
(37.2%) and $G$ (34.6%), followed by $SD$ (27.9%). As with snow albedo reduction,
$R_{eff}$ plays only a minor role (0.2%). Our data indicate that spatial variability in
radiative forcing is highly dependent on $G$, a pattern that we attribute to the high degree
of variability in latitude-dependent solar radiation among ISCA.




On a regional scale, the respective contributions of LAP content, $G$, and $SD$ are also
comparable among the four study areas, accounting for 27.4%, 37%, and 34% of
radiative forcing in NEC, 24.9%, 38%, and 36.8% in NA, and 35.6%, 35.2%, and 27.2%
in EUA. The Arctic radiative forcing is dominated by LAPs (52.8%) and $G$ (46.4%).
In summary, LAPs play a dominant role in the spatial variability of snow albedo
reduction and radiative forcing. Our results also highlight the significant contribution
of $SD$ to snow albedo reduction and $G$ to radiative forcing.
**4.6. Comparisons with model simulations**
To investigate the global distribution and variance of RFLS, previous studies have
tended to rely on Earth system models with minimal cross-checking from *in situ*
measurements or remote sensing observations (Qian et al., 2015; Skiles et al., 2018). In
this study, we compared MODIS retrievals with two model-based estimates to improve
our understanding of the magnitude of RFLS on a global scale. Flanner et al. (2009)
simulated springtime RFLS for the Northern Hemisphere by applying a Global Climate
Model (GCM) coupled with a SNICAR model, the results of which are presented in
Fig. 8a and Fig. S5a.
Figure 8b depicts the direct comparison of our MODIS retrievals ($RF_{MODIS,daily}^{LAPs}$) to the
simulations of Flanner et al. (2009) ($RF_{GCM}$). In general, $RF_{MODIS,daily}^{LAPs}$ and $RF_{GCM}$
are of the same magnitude and both fall in the range 0.10–4.6 W m$^{-2}$. The MAE for





$RF_{GCM}$ against $RF_{MODIS,daily}^{LAPs}$ is 1.6 W m$^{-2}$. Regionally, the average $RF_{MODIS,daily}^{LAPs}$ in
NEC (NA) is 11 W m$^{-2}$ (1.4 W m$^{-2}$), a value that is higher by a factor of 2.5 (2.0) than
the $RF_{GCM}$ value of 4.2 W m$^{-2}$ (0.69 W m$^{-2}$). In contrast, $RF_{MODIS,daily}^{LAPs}$ (0.29 W m$^{-2}$)
for the Arctic is systematically lower than $RF_{GCM}$ value (1.1 W m$^{-2}$) by a factor of 3.9.
For EUA, average $RF_{MODIS,daily}^{LAPs}$ (1.5 W m$^{-2}$) and $RF_{GCM}$ (1.9 W m$^{-2}$) are comparable,
while correlations between $RF_{MODIS,daily}^{LAPs}$ and $RF_{GCM}$ for NA, NEC, and the Arctic
are all significant at the 99 % confidence level (R$^2$ = 0.36, 0.43, and 0.72, respectively).
Employing ensemble-average snow BC concentrations from CESM2 and CESM2-
WACCM, we also calculated January–February radiative forcing ($RF_{CMIP6}$) for the
Northern Hemisphere ISCA during the period 2003–2014 (Fig. 9a). Statistics are
presented in Fig. S5b. Briefly, $RF_{CMIP6}$ exhibits strong spatial inhomogeneity, with
values ranging from 0.068 W m$^{-2}$ to 4.6 W m$^{-2}$. The highest regional average in
$RF_{CMIP6}$ occurs in NEC ($\geq$5 W m$^{-2}$) and the lowest in the Arctic ($\leq$0.2 W m$^{-2}$),
consistent with $RF_{MODIS,daily}^{LAPs}$.
Figure 9b depicts the comparison of $RF_{MODIS,daily}^{LAPs}$ and $RF_{CMIP6}$, for which the MAE
of $RF_{CMIP6}$ to $RF_{MODIS,daily}^{LAPs}$ is 0.90 W m$^{-2}$. In NEC, $RF_{CMIP6}$ (8.4 W m$^{-2}$) compares
well with $RF_{MODIS,daily}^{LAPs}$ (10.60 W m$^{-2}$), with a significant correlation at the 99%
confidence level (R$^2$ = 0.43). For NA and EUA, $RF_{CMIP6}$ (0.69 W m$^{-2}$ and 1.6 W m$^{-2}$,
respectively) are lower than $RF_{MODIS,daily}^{LAPs}$ (1.4 W m$^{-2}$ and 1.5 W m$^{-2}$, respectively)
and the spatial correlations between them are poor. In the Arctic, $RF_{CMIP6}$ is
significantly correlated with $RF_{MODIS,daily}^{LAPs}$ at the 99% confidence level (R$^2$ = 0.80).



However, $RF_{CMIP6}$ (0.15 W m$^{-2}$) is lower than $RF_{MODIS,daily}^{LAPs}$ (0.29 W m$^{-2}$) by a factor
of 2, an outcome that is contrary to the comparison between $RF_{GCM}$ and $RF_{MODIS,daily}^{LAPs}$.
Overall, the RFLS derived from our MODIS retrievals and modeling-based estimates
exhibit a same magnitude over the Northern Hemisphere. In NEC, the MODIS- and
model-derived estimates show good general agreement, indicating the satisfactory
performance of Earth system modeling in this heavily polluted region. In NA and EUA,
average radiative forcing values are comparable but the spatial correlation is relatively
poor, while MODIS retrievals for the Arctic are significantly correlated with those
simulations.
**5.   Discussion**
In recent decades, there has been increasing scientific interest in snow LAPs due to
their role in the climate system, and numerous studies have attempted to evaluate RFLS.
In addition to making global-scale comparisons between our MODIS retrievals and
model-based estimates, this study collects a comprehensive set of radiative forcing
estimates, based on local-scale observations and remote sensing, to make quantitative
regional- and global-scale comparisons and synthetically evaluate the magnitude of
RFLS (Fig. 10). This approach also affords the opportunity to examine the MODIS
retrievals used in our study.
For instantaneous RFLS, Pu et al. (2019) reported an average value for NEC of ~45





W m$^{-2}$, based on high-resolution (500 m) MODIS retrievals (Fig. 10a), which compares
well with our findings (24–53 W m$^{-2}$). This agreement indicates that the coarse-
resolution (0.05°) MODIS data used in this study are sufficient to establishing RFLS.
In NA, Painter et al. (2012a) reported RFLS values of ~100–250 W m$^{-2}$ for the Upper
Colorado River Basin, while Seidel et al. (2016) obtained values of 20–200 W m$^{-2}$ in
the Sierra Nevada and Rocky Mountain (Fig. 10a). For EUA, Di Mauro et al. (2015)
estimated a maximum RFLS of 153 W m$^{-2}$ in the European Alps (Fig. 10a). In the Arctic,
Nagorski et al. (2019) reported a maximum RFLS for Alaska of >4 W m$^{-2}$ in May and
87 W m$^{-2}$ in June (Fig. 10a), while Ganey et al. (2017) provided a microbe-LAP-
induced noontime radiative forcing of >~20 W m$^{-2}$ (Fig. 10a). We note that the findings
of previous studies are approximately one order of magnitude higher than our results in
NA, EUA, and the Arctic. We attribute this partly to the fact that those earlier studies
focused on a local-scale estimates for sites with high LAP loadings, whereas our study
provides regional-scale estimates.
Relative to estimates of instantaneous RFLS, which are comparatively rare, average-
daily RFLS has been paid more attention climatologically. Figure 10b shows the
calculated average-daily RFLS based on *in situ* measurements and remote sensing.
Dang et al. (2017) reported RFLS values of 7–18 W m$^{-2}$, 0.6–1.9 W m$^{-2}$, and 0.1–0.8
W m$^{-2}$ for northern China, North America, and the Arctic, respectively, which we note
are comparable with our own retrievals. In NA, Sterle et al. (2013) estimated a daily-
averaged RFLS of ~5 W m$^{-2}$ for the eastern Sierra Nevada, while Miller et al. (2016)





reported a daily RFLS of <4 W m$^{-2}$ in the San Juan Mountains. Both values are higher
than our estimate (~1.4 W m$^{-2}$), potentially due to the significant dust deposition in
those areas.
Figure 10c shows the average-daily RFLS simulated by regional and/or global climate
models. For NEC, Zhao et al. (2014) and Qian et al. (2014) reported values of 10
W m$^{-2}$ and 5–10 W m$^{-2}$, respectively. In NA, the latter study also provided an estimate
of 2–7 W m$^{-2}$ for the central Rockies and southern Alberta (Qian et al., 2014), while
Oaida et al. (2015) reported an average RFLS of 16 W m$^{-2}$ over the western US. In
EUA, Flanner et al. (2014) concluded that a RFLS of 0.1–1 W m$^{-2}$ was caused by the
deposition of volcanic ash in snow. Finally, Qian et al. (2014) and Qi et al. (2017)
estimated RFLS values of <0.3 W m$^{-2}$ and 0.024–0.39 W m$^{-2}$ for the Arctic, respectively.
We consider our retrievals for NEC, EUA, and the Arctic to be comparable with these
regional model simulations, despite some disparity. However, we note that our result
for NA is significantly lower than those of previous studies.
For the North Hemisphere as a whole, Bond et al. (2013) estimated a climate forcing of
0.13 W m$^{-2}$, while Hansen and Nazarenko (2004) and Wang et al. (2014a) reported
RFLS values of 0.3 W m$^{-2}$ and 0.45 W m$^{-2}$, respectively. Each of these previous values
is significantly lower than our retrieval (~1.7 W m$^{-2}$). We attribute this disparity to the
inclusion in those studies of low-LAP boreal forests, which contribute very little to
overall radiative forcing yet exhibit a high degree of uncertainty. Skiles et al. (2018)
also concluded that modeled RFLS might be biased if the snow-covered area itself is





not accurately represented.
Overall, we consider our MODIS-based retrievals to be physical realistic on both
regional and global scales, although we note a number of differences between our
results and those generated by different methods. On the other hand, while *in situ*
measurements are the most precise, their spatial coverage is restricted by logistical
limitations and the extreme environments involved. Conversely, models can provide
broad perspectives of climatic impacts yet are typically undermined by large uncertainty.
Therefore, we argue that remote sensing provides a powerful technique, with high
spatial and temporal resolutions, that can bridge the gap between *in situ* measurements
and climate models and reduce the uncertainties associated with the latter. Further
retrieval of remote-sensing data, including the use of multiple satellites and sensors, is
therefore warranted to exploit this opportunity fully. We also indicate the fact that parts
of central EUA, such as Middle East, are characterized by high dust deposition,
however, studies are barely performed but desired. Finally, we note that *in situ*
observations remain limited, and more field campaigns are needed to constrain remote
sensing retrievals and modeling simulations.
**6. Conclusion**
We presented a global-scale evaluation of the radiative forcing of LAPs in the Northern
Hemisphere snowpack (RFLS), estimated from remote-sensing data. The satellite-
retrieved RFLS also has implications for expanding the value of limited *in situ*





measurements, which can provide valuable information for climate models and help
optimize model simulations.
Based on the corrected snow albedo reduction ($\Delta\alpha_{MODIS,corrected}^{LAPs}$), we used the
SBDART model to calculated instantaneous RFLS ($RF_{MODIS,ins}^{LAPs}$) and average-daily
RFLS ($RF_{MODIS,daily}^{LAPs}$) during January–February for the period 2003–2018. For the
Northern Hemisphere as a whole, average $\Delta\alpha_{MODIS,corrected}^{LAPs}$ is ~0.0246, $RF_{MODIS,ins}^{LAPs}$
is ~5.9 W m$^{-2}$, and $RF_{MODIS,daily}^{LAPs}$ is ~1.7 W m$^{-2}$. We also observed distinct spatial
variability in snow albedo reduction and RFLS. The highest regional-average
$\Delta\alpha_{MODIS,corrected}^{LAPs}$ (~0.1669), $RF_{MODIS,daily}^{LAPs}$ (~11 W m$^{-2}$), and $RF_{MODIS,ins}^{LAPs}$ (~36
W m$^{-2}$) occur in northeastern China, while the lowest regional averages of ~0.0123,
~0.29 W m$^{-2}$, and ~1.1 W m$^{-2}$, respectively, are observed in the Arctic.
Following this assessment, we made quantitative attributions of the spatial variability
in snow albedo reduction and radiative forcing. Our results indicate that the LAP
content is the largest contributor (57.6%) to spatial variance in snow albedo reduction,
followed by snow depth (38.1%), whereas snow grain size (2.7%) and the geographic
factor $G$ (1.6%) are only minor contributors on a Northern Hemispheric scale. LAP
content and $G$ account for 37.2% and 34.6% of the spatial variability of radiative
forcing, respectively, following by $SD$ (27.9%) over Northern Hemisphere.
Retrieved RFLS values are compared spatially with the model-derived estimates of the
Global Climate Model (GCM) and the Coupled Model Intercomparison Project (CMIP
Phase 6). Our results indicate that MODIS retrievals provide show the same magnitude



with modeled estimates for Northern Hemisphere. However, although the Earth system
models perform well in NEC, there remain large uncertainties in the Arctic. To evaluate
and examine the MODIS retrievals synthetically, we then compared the retrieved RFLS
to previously published estimates, including local-scale observations, remote sensing
retrievals, and regional- and global-scale model simulations. The results of this
evaluation suggest that MODIS retrievals are generally realistic, despite a number of
important differences among the various methods.
Finally, we urge the community to expand the ground-based measurements of the global
snowpack, particularly in those regions currently lacking *in situ* observations. Such
development would help further constrain and improve satellite-based retrievals in the
future. We propose that climate models incorporating these refined remote sensing
retrievals should be able to capture the RFLS more accurately, thereby providing more
reliable estimates of the future impacts of global climate change.
**Data availability.**
MODIS data can be found at https://earthdata.nasa.gov/ (last access: 20 January 2019).
CERES data can be found from NASA's Clouds and the Earth's Radiant Energy System
at https://ceres.larc.nasa.gov (last access: 12 April 2019). Shuttle Radar Topography
Mission (SRTM) digital elevation data are provided by the US Geological Survey at
https://www.usgs.gov/ (last access: 9 December 2018). Snow depth can be found from
ERA-Interim at https://www.ecmwf.int (last access: 15 January 2019). BC emission



data can be found at http://inventory.pku.edu.cn (last access: 5 June 2019). BC
deposition data can be found at https://gmao.gsfc.nasa.gov/reanalysis/MERRA-2/ (last
access: 5 June 2019). CMIP6 data can be found at https://esgf-node.llnl.gov/ (last access:
15 July 2019). Surface measurement datasets are from Wang et al. (2013, 2017), Ye et
al. (2012) and Doherty et al. (2010, 2014). Springtime radiative forcing due to LAPs in
snow is derived from a GCM run by Flanner et al. (2007).
**Author contributions.**
PW and WX designed the study and evolved the overarching research goals and aims.
CJC carried the study out and wrote the first draft with contributions from all co-authors.
CJC and STL applied formal techniques such as statistical, mathematical and
computational to analyze study data. ZY prepared input data and managed activities to
annotate, scrub data and maintain research data. WDY completed the implementation
of the computer code and supporting algorithms used for the calculations in this study.
PW and WX assumed oversight and leadership responsibility for the research activity
planning and execution. All authors contributed to the improvement of results and
revised the final paper.
**Competing interests.**
The authors declare that they have no conflict of interest.
**Acknowledgments**
This research was supported jointly by the National Key R&D Program of China
(2019YFA0606800), the National Natural Science Foundation of China (grants





41975157, 41775144, and 41875091).

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





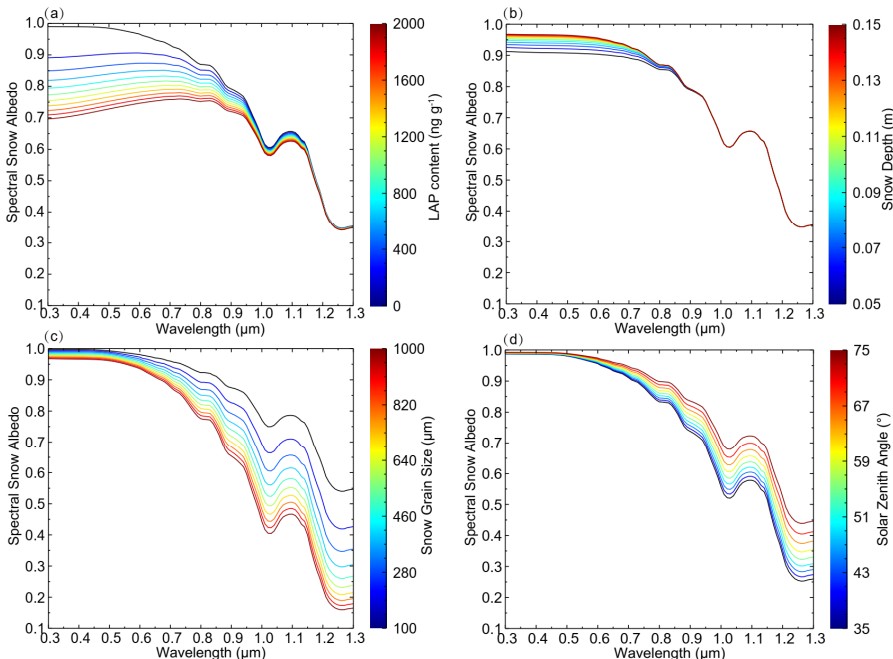

Figure 1. Variations in spectral snow albedo due to (a) LAP content (ng g$^{-1}$), (b) snow depth (m), (c) snow grain size (μm), and (d) solar zenith angel (deg.).




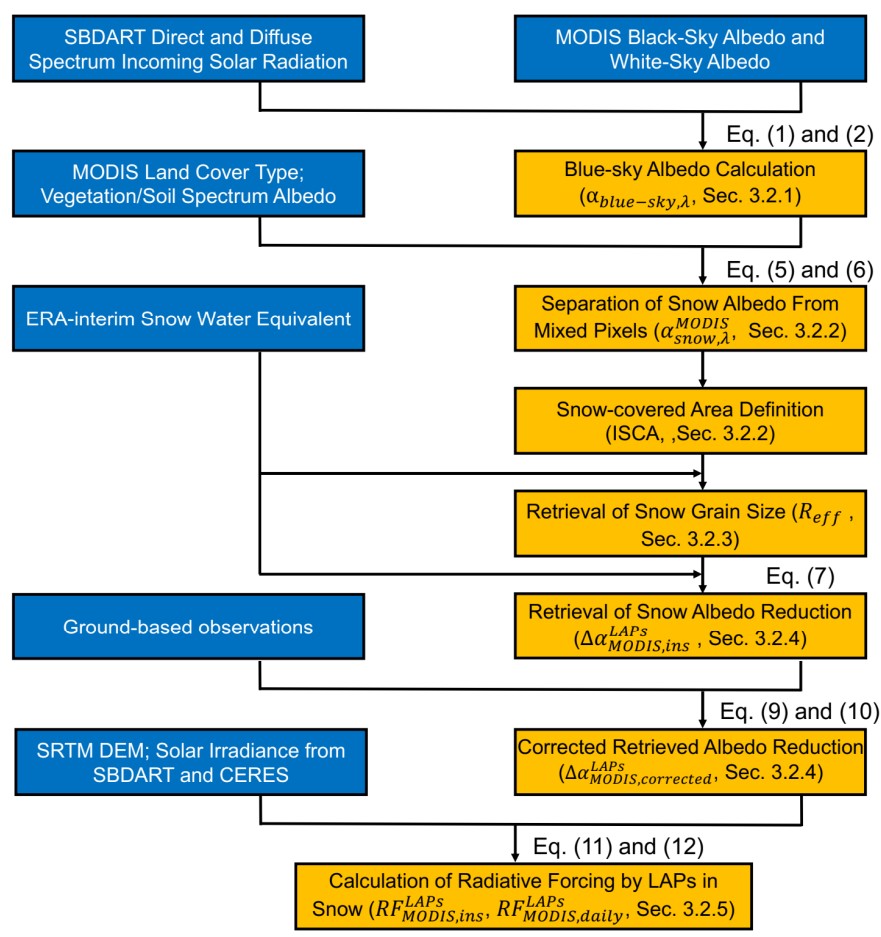

2 Figure 2. Workflow depicting the calculation and validation of radiative forcing of LAPs in snow:

3 the blue boxes denote the external input data, while the orange boxes are used for calculations in

4 this study.

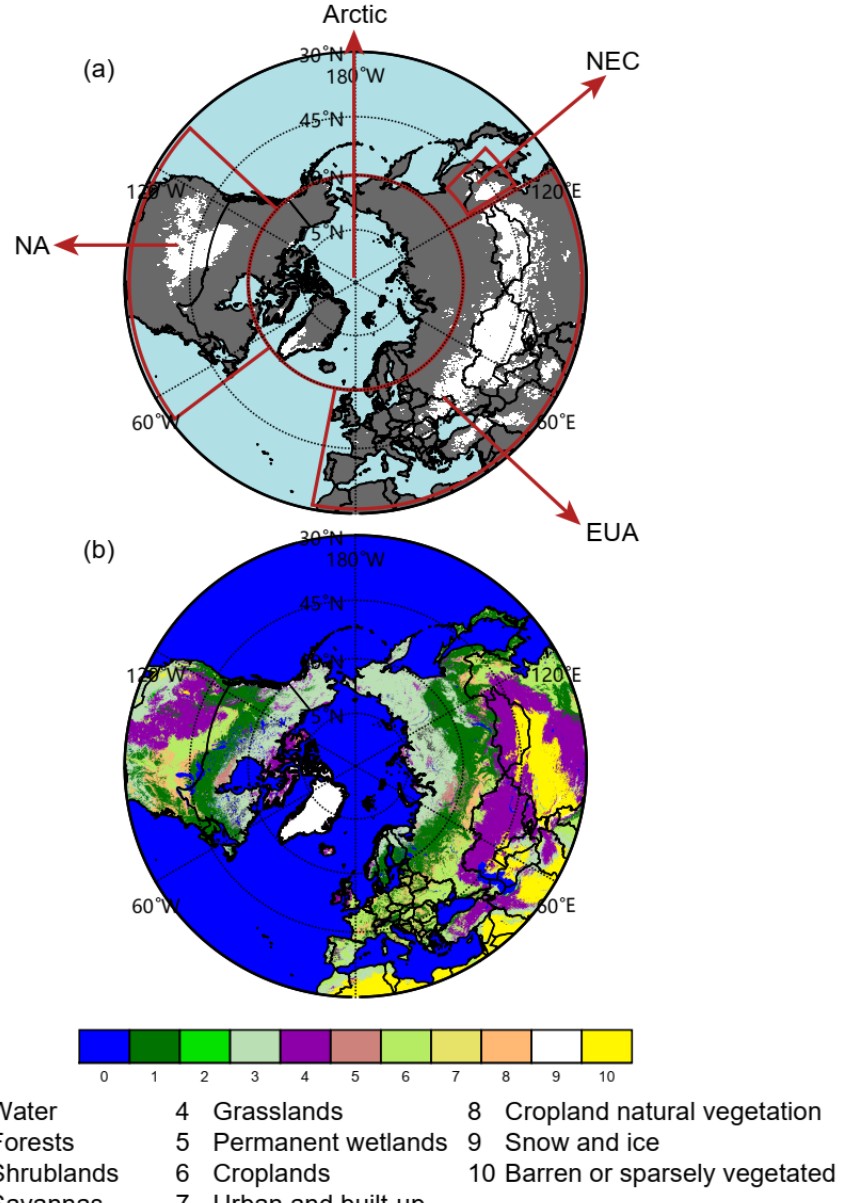

| 0 | Water | 4 | Grasslands | 8 | Cropland natural vegetation |
| 1 | Forests | 5 | Permanent wetlands | 9 | Snow and ice |
| 2 | Shrublands | 6 | Croplands | 10 | Barren or sparsely vegetated |
| 3 | Savannas | 7 | Urban and built-up | | |

2  Figure 3. Spatial distributions of (a) identified snow-covered areas (ISCA) and (b) the different land-

3  cover types, based on MODIS data, for the Northern Hemisphere. ISCA (white) can be separated

4  into northeastern China (NEC), Eurasia (EUA), North America (NA), and the Arctic.

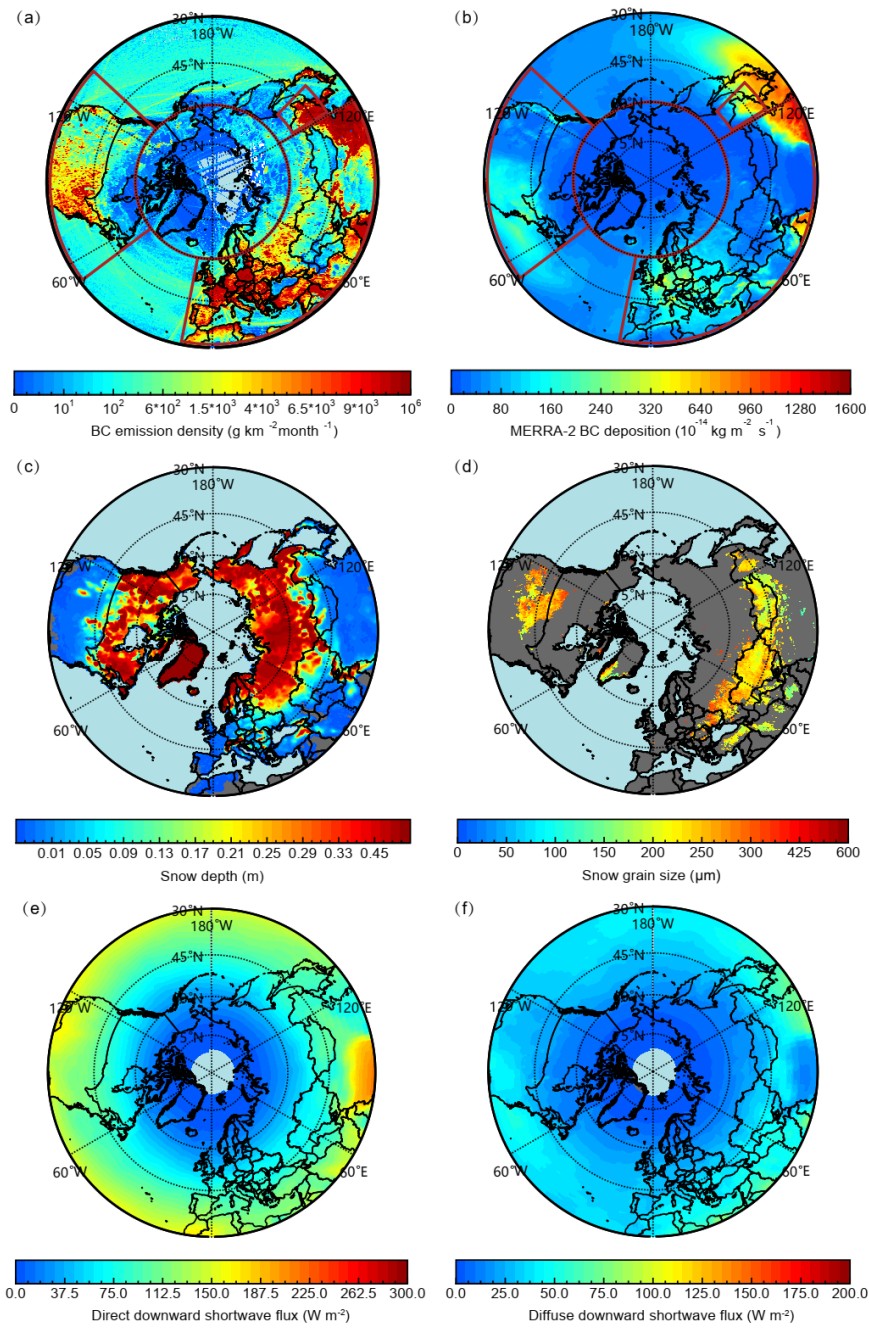

2 Figure 4. Spatial distributions of average (a) BC emissions density, (b) BC deposition from

3 MERRA-2, (c) snow depth from ERA-interim, (d) snow grain size retrieved by MODIS, (e) direct

4 and (f) diffuse solar irradiance at the surface in January–February from CERES. BC emissions



density is for the period 2003–2014 and employs data from the research group at Peking University,
with additional data collected between 2003 and 2018.

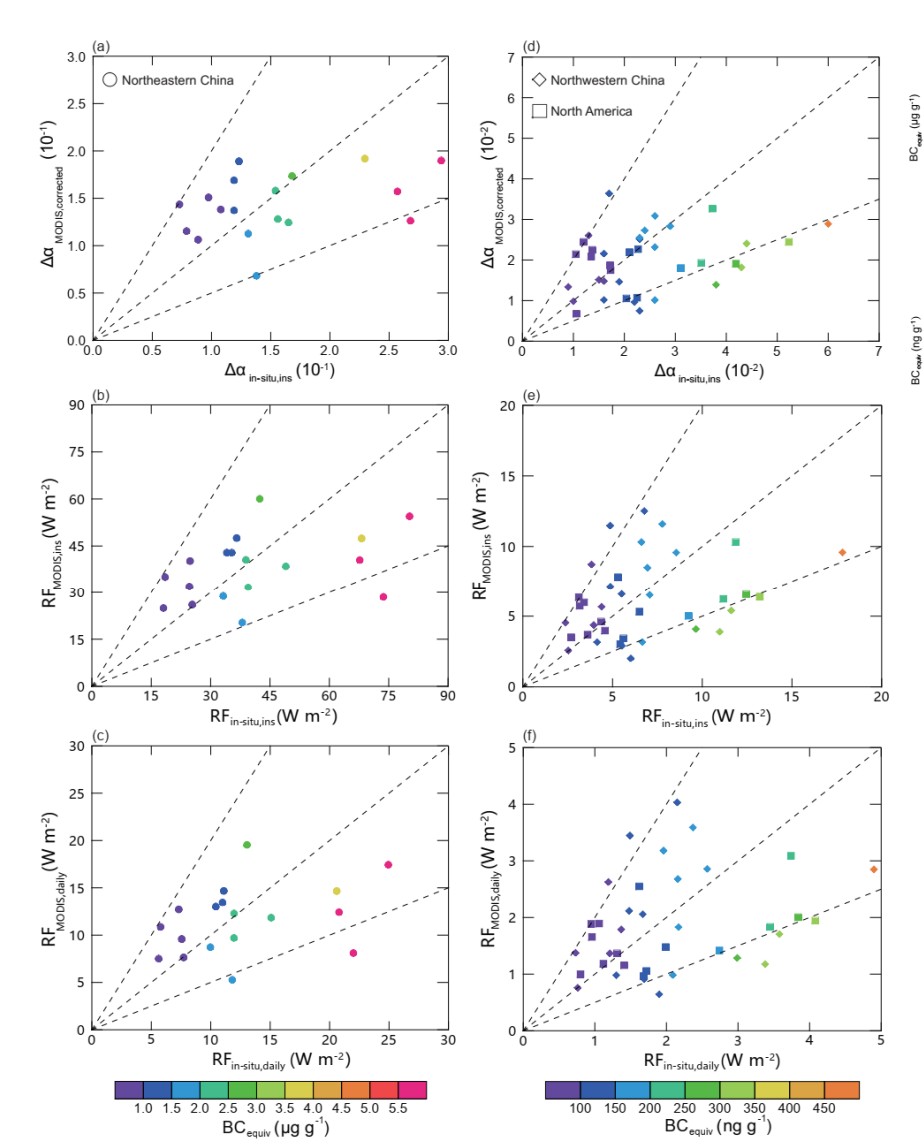

Figure 5. Scatterplots of (a) $\Delta\alpha_{MODIS,corrected}$ versus $\Delta\alpha_{in-situ,ins}$, (b) $RF_{MODIS,ins}$ versus
$RF_{in-situ,ins}$, and (c) $RF_{MODIS,daily}$ versus $RF_{in-situ,daily}$ in heavily polluted areas. Panels (d)–
(f) illustrate the same scatterplots as in (a)–(c) but for slightly polluted regions. Circles, diamonds,
and squares represent the snow samples collected in NEC, NWC, and NA, respectively.



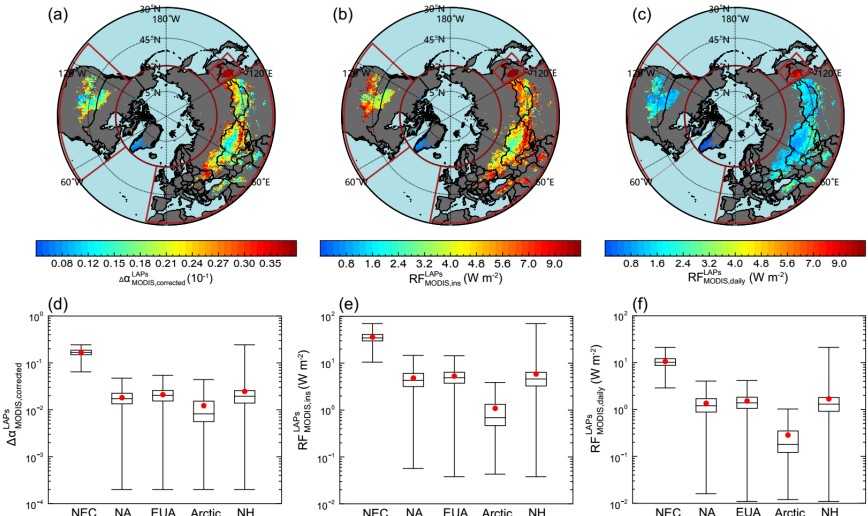

Figure 6. Spatial distributions of averaged (a) $\Delta\alpha_{MODIS,corrected}$ , (b) $RF_{MODIS,ins}$ , and (c) $RF_{MODIS,daily}$ and statistics for regionally averaged (d) $\Delta\alpha_{MODIS,corrected}$, (e) $RF_{MODIS,ins}$, and (f) $RF_{MODIS,daily}$ for the Northern Hemisphere in January–February during the period 2003–2018. The boxes denote the 25th and 75th quantiles, and the horizontal lines represent the 50th quantiles (medians), the averages are shown as red dots; the whiskers denote the 5th and 95th quantiles.



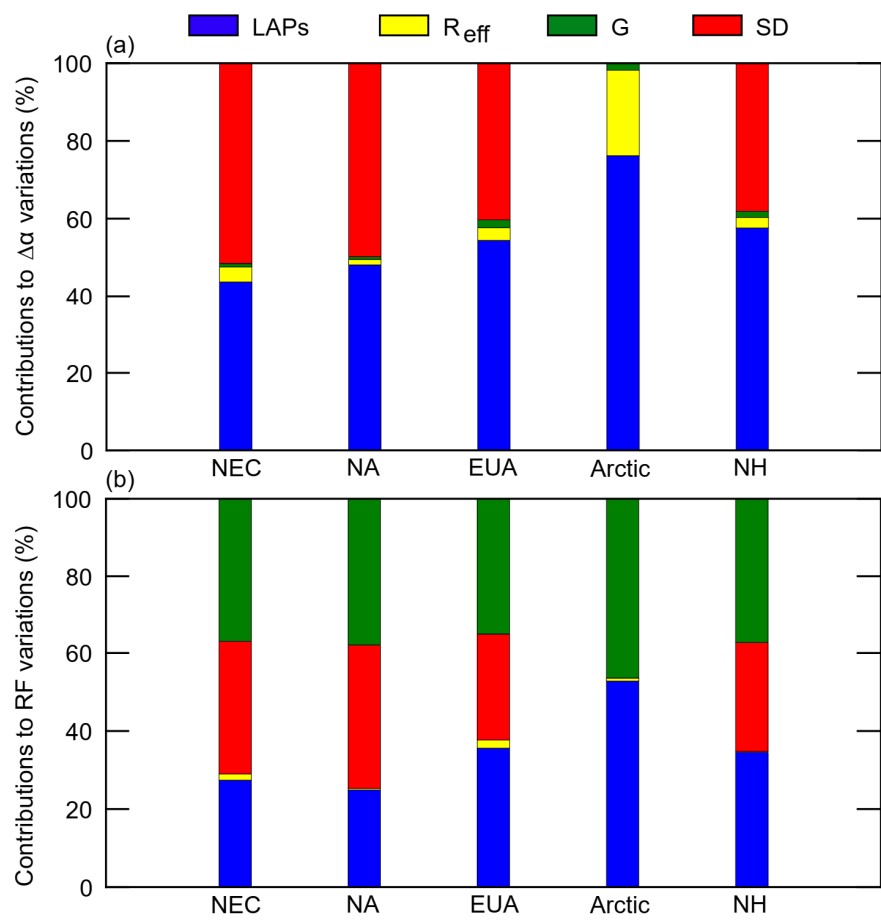

Figure 7. Fractional contributions of LAPs, snow grain size ($R_{eff}$), geographic factor ($G$), and snow depth ($SD$) to the spatial variations of (a) snow albedo reduction and (b) radiative forcing.



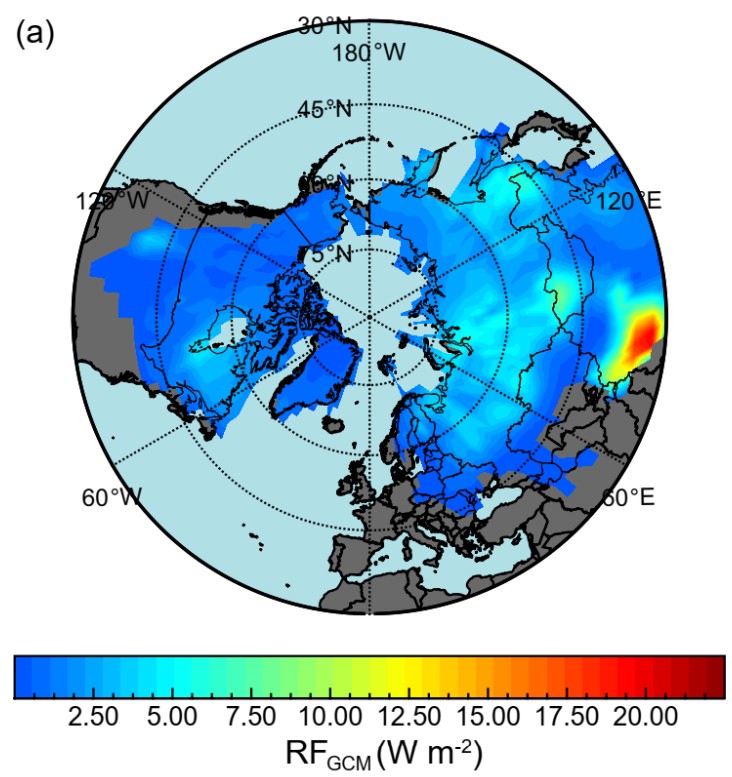

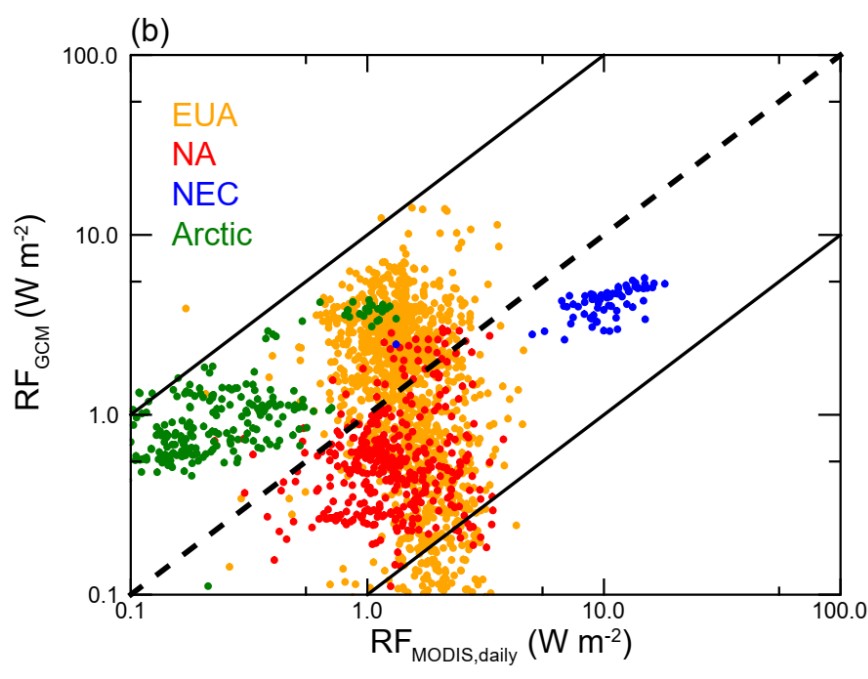



1    Figure 8. Spatial distributions of (a) springtime radiative forcing ($RF_{GCM}$) due to LAPs in snow,

2    derived from a GCM run by Flanner et al. (2007), and scatterplot of (b) $RF_{MODIS,daily}$ versus

3    $RF_{GCM}$.



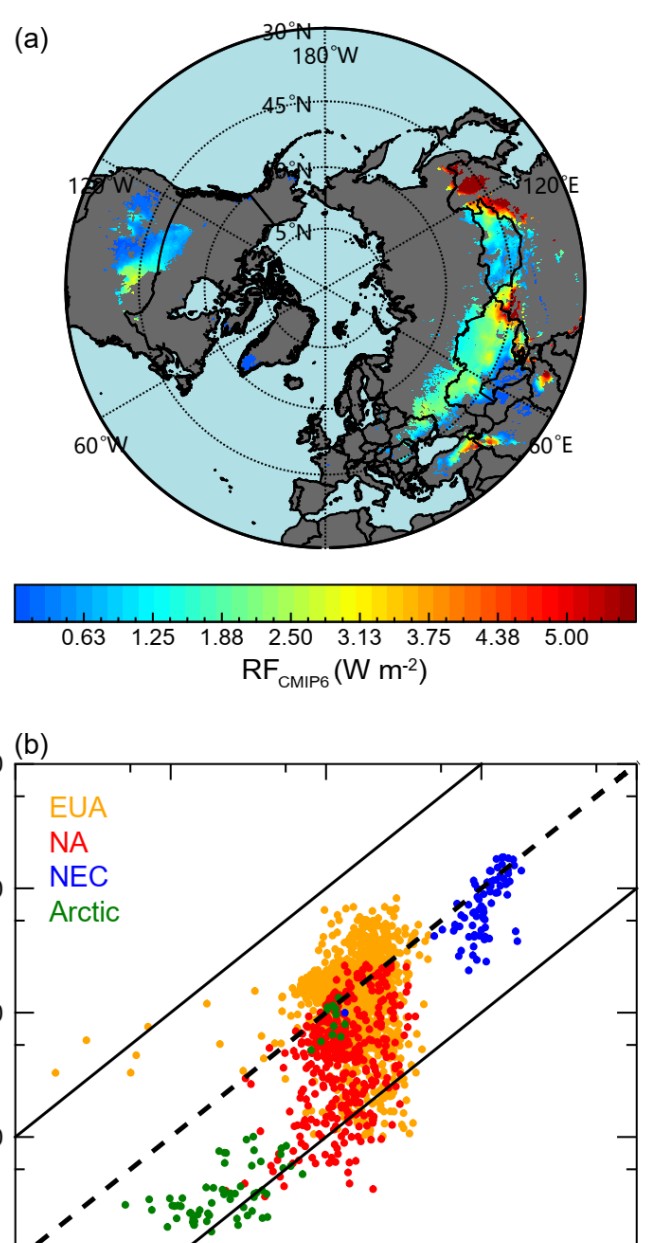

Figure 9. (a) Spatial distributions of average-daily radiative forcing ($RF_{CMIP6}$), based on the CMIP6





ensemble-average soot content of snow in January–February for the period 2003–2014. (b)
Scatterplot of $RF_{MODIS,daily}$ versus $RF_{CMIP6}$.



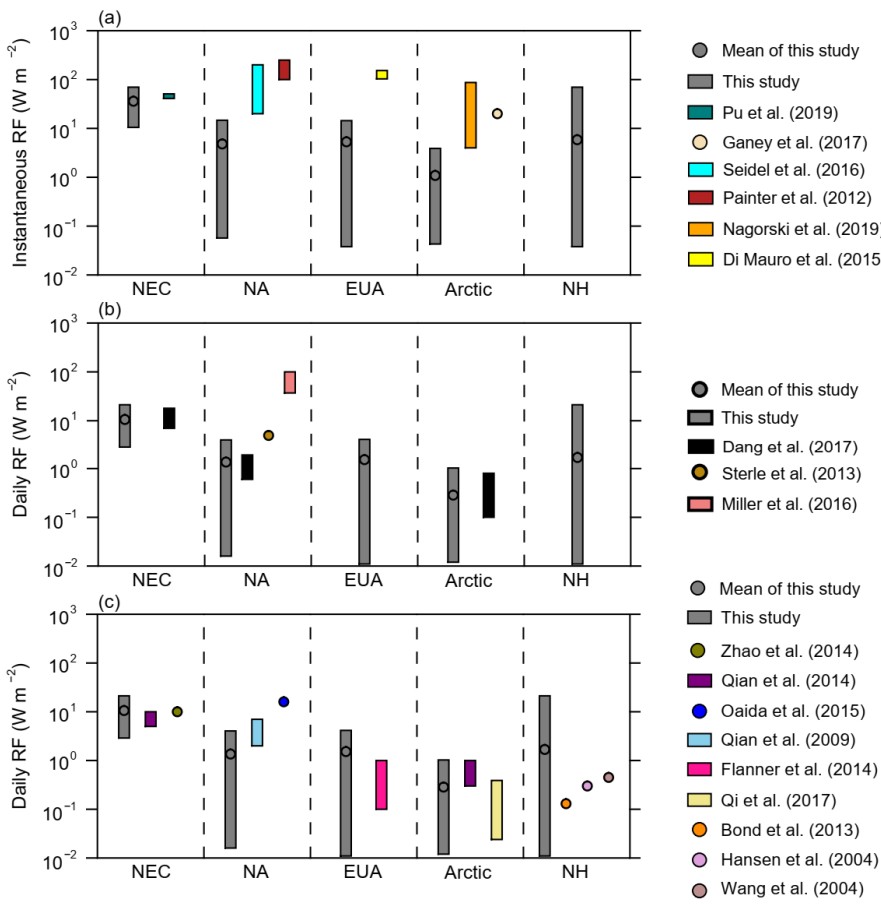

Figure 10. Comparisons of radiative forcing due to LAPs in snow (this study) with observed and
model-simulated values from previous studies.

