# Peer review of "Satellite-based radiative forcing by light-absorbing particles in snow across the Northern Hemisphere"

_Atmospheric Chemistry and Physics, 2020_

## Referee Comment (RC1) · Anonymous Referee #1 · 13 May 2020

**General comments**

This paper presents a method for estimating the radiative forcing due to light-absorbing particles (LAPs) in snow (RFLS) using several data sources, which include MODIS albedos, snow grain size derived from MODIS data, snow depth from the ERA-Interim reanalysis, surface downwelling solar radiation from CERES, and finally, in situ measurements of BC in snow (used for computing correction factors for the algorithm). The proposed approach allows the estimation of RFLS in larger areas than would be possible with in situ measurements alone. It thus provides an additional data source complementing estimates from in situ data and climate models. As noted in the intro-

duction, there are previous studies that utilized MODIS to retrieve the radiative forcing of LAPs in snow, but this might be the first one to consider the spatial variability in RFLS between different regions. The approach is further employed to analyze the factors underlying the spatial variation of RFLS, finding that the variations in LAP content, snow depth and geographical factors (e.g., latitude) are more important than those in snow grain size (Fig. 7). Furthermore the retrieved values of RFLS are compared with results from a few climate models (Figs. 8 and 9) and with previous studies (Fig. 10).

A practical limitation of the proposed approach is that it can only be applied in regions with no/very short vegetation. Also, judging by the correction factors needed to eliminate systematic differences to RFLS derived from in situ data, it appears that the approach works fairly well in heavily polluted regions, but for regions with relatively clean snow, the uncertainties are very large (Fig. S2b). So if one interprets "hemisphere-scale" values (p. 6, line 17) as "hemisphere-mean" values, they cannot yet be obtained with this approach.

There is certainly enough new material in this work to be published in ACP. The paper is reasonable well written especially as regards the description of the approach, but I think there are disturbingly many numerical values in the text towards the end, and possibly some apples-to-oranges comparisons.

**Specific comments**

1. It is not justified to "sell" the values averaged over all ISCAs as the Nothern Hemisphere (NH) mean values (e.g., p.2, lines 11-13, and p. 34, lines 6-7) since they really represent only a small part of the NH land area. The approach samples only areas with (nearly) full snow cover and no/very short vegation, which naturally results in a high bias in the computed "NH average" RF. The assumption of clear-sky conditions

further increases the RF values, while the analysis of only January and February data decreases the RF in the Arctic, but perhaps increases it at midlatitudes, compared to annual-mean values. In general, you should avoid listing numerical values without explaining what they really mean, especially in the abstract.

2. Specifically, the abstract should state that these are clear-sky values, that the albedo reduction refers to wavelengths 300-1300 nm, and the RF values refer to areas with full snow cover and little/no vegetation above snow.

3. p. 6, lines 14-16, Section 2.4, and Section 4.6: Taking only two CMIP6 models, and calling them "CMIP6" or "CMIP6 ensemble mean" is misleading, especially as the two models (CESM2 and CESM2-WACCM) are very closely related and produce nearly identical results (Fig. S4). It would be advantageous to use data from more CMIP6 models, if data from more models has now become available. If not, just take CESM2 and call it CESM2!

In addition, instead of a "Global climate model", you should use the specific model name for Flanner et al. (2009), that is CAM3.1. Incidentally, it is a precedessor of the atmospheric and land components of CESM2.

4. p. 7, line 6 and elsewhere: Why do you only use data for January and February? The reason for this should be stated explicitly. Perhaps because the midlatitude snow cover is most extensive then? However, this choice screens out almost all of the Arctic, due to the low sun angles, so that the "Arctic" RFLS values is this work in practice only represent southern Greenland. Also, considering spring months would increase the Arctic RFLS values substantially.

5. p. 9. A brief description of the in situ BC measurements employed to correct the RFLS values should be included in Section 2 (at least, regions and references), perhaps between current Sections 2.2 and 2.3. Do these measurements represent BC or LAPs in general?

6. p. 9, lines 7–10: What was the reason for converting SWE to snow depth? To my knowledge, this has no effect on the results (in the end, SNICAR cares of SWE only).

7. p. 9, Section 2.3. It should be stated how/why these emission data were used. I get the impression that they were used just as background information (not in estimating the RFLS).

8. p. 11, lines 11-16: You describe how SBDART has several options for defining the atmospheric properties. It would be more important to tell what was assumed in the present calculations (also regarding aerosols).

9. p. 12, line 11: You could add snow grain shape to this list.

10. p. 12, lines 20-21: "previous studies have tended to assume a semi-infinite snow-pack". This is a good point, and I think it would be worth showing how much this influences the results. Consider adding a figure which shows the ratio of RFLS computed using the actual (ERA-Interim) snow depth vs. RFLS computed using semi-infinite snow.

11. p. 13, line 20: add "...for clear-sky conditions" at the end of the sentence.

12. p. 16, Eq. (7): Please state explicitly that the impact of LAPs on snow albedo computed in this work refers to the spectral range 300-1300 nm only. There is a chance of misinterpretation here, as usually people think of broadband albedo integrated over the

entire downwelling solar spectrum at the surface. (An alternative would be to calculate "real" broadband albedo changes, integrated over 0.3–4 $\mu$m or at least 0.3–2.5 $\mu$m). This choice should not matter for RFLS, however.

13. In Eq. (2), diffuse and direct spectral solar radiation are added as such $(E_{\mathrm{dif}}(\lambda; \phi) + E_{\mathrm{dir}}(\lambda; \phi))$, suggesting that they both are defined wrt. a horizontal surface, but in Eq. (7) (and Eqs. (10) and (11)) the direct radiation is weighted by the cosine of local solar zenith angle ($E_{\mathrm{dir},\lambda} \cos \beta + E_{\mathrm{dif},\lambda}$), which implies that the direct radiation is defined wrt. a surface perpendicular to Sun's direction. This seems inconsistent.

14. p. 17, line 7: "we assumed that the properties for snow and LAPs remain invariable throughout the day". In fact, if you keep the snow physical properties and LAP concentration constant, the impact of LAPs on snow albedo decreases with increasing solar zenith angle, so the use of $\Delta\alpha_{\mathrm{MODIS,corrected}}^{\mathrm{LAPs}}$ evaluated at noon probably overestimates the daily-average impact of LAPs somewhat. (I would guess, perhaps of the order of 10%, but this is something that you could check with SNICAR.)

15. In Eq. (11), is $\cos\beta$ the daytime mean value?

16. p. 22, lines 16-17: You should remind the reader that this result refers specifically to the months of January and February. In spring and early summer, much of the Arctic is still snow-covered and solar radiation is much more abundant, so RFLS is substantially larger than in January-February.

17. p. 22, line 20. "In situ observations of snow albedo reduction" actually refer to the albedo reduction calculated using in-situ observed LAPs. Here, it should be noted what was the measure of LAPs used in the in situ observations? Was it BC (excluding dust)

[Figure]

or equivalent BC (implicitly also including dust). I guess in-situ obervations usually yield the latter?

18. p. 22, line 11. These corrections deserve a bit more discussion. The value $c_{polluted} = 1.1$ suggests that the approach works rather well for heavily polluted snow. However, the value $c_{clean} = 5.6$ for "relatively pure" snow, along with the scatter of points in Fig. S2, suggests that the method becomes quite inaccurate then. Can you comment on the possible reasons for that? Perhaps the limiting factor is simply the accuracy of albedo calculations and observations, and a possible systematic bias between the two? For example, for 100 ng/g of BC (which many would already consider not so clean snow!) the albedo reduction is only $\sim$0.02. So, if in Eq. (7) $\alpha_{snow,\lambda}^{mdl}$ is biased high and/or $\alpha_{snow,\lambda}^{MODIS}$ is biased low, this would result in $c > 1$, the more so the cleaner the snow.

19. p. 23–26: I think the large number of numerical values in the text is disrupting to the reader. Some concrete suggestions would be: 1) for p. 23, lines 14-20 provide the MAE and RMS statistics in the figure panels in Fig. 5, 2) in Section 4.4., put the numerical values in a table. If you prefer to keep them in the text, you could at least skip the instantaneous RF values.

20. p. 28, line 10 – p. 29, line 7: As noted above, the model used by Flanner et al. (2009) should be called "CAM3.1" rather than "GCM". More importantly, you discuss springtime RF for Flanner et al. Did you compute springtime values for the MODIS retrievals too? This should be made clear in the text. Comparing January–February values with springtime (March–May?) values would be meaningless.

21. p. 31-32 and Fig. 10. The comparison with previous radiative forcing estimates is interesting, but one should be careful not to compare apples with oranges let alone

watermelons – or at least be explicit about when this is being done. In other words, I think you should provide more information about the previous studies considered here. The RF differences could arise from the consideration of different regions, different seasons, clear-sky vs. all-sky forcing etc., so these details should be mentioned. This information would probably best fit in a table.

22. p. 31, line 21 – p. 32, line 1: "Miller et al. (2016) reported a daily RFLS of $< 4$ $\mathrm{W\,m}^{-2}$". Figure 10b (2nd panel) shows much larger values.

23. p. 32, line 7: Should this be Qian et al. (2014) or Qian et al. (2009) (cf. Fig 10c, second panel).

24. p. 32, lines 16-17: It is stated that Wang et al. (2014a) reported a northern hemisphere RFLS value of 0.45 $\mathrm{W\,m}^{-2}$. However, so far I can tell, that paper is concerned with the direct radiative forcing due to BC in air (not snow). Furthermore, Fig. 10c refers to Wang et al. (2004), which is not present in the reference list.

25. p. 32, lines 15–21. I think your explanation is in principle correct, although at least the values of Bond et al. (2013) and Hansen and Nazarenko (2004) are annual-mean values, not January-February. But the fundamental point here is that your approach cannot provide northern-hemisphere (NH) mean values, which the cited studies attempt to provide, uncertainties notwithstanding. It can only provide values for ISCAs that are snow-covered and without much vegetation. For true NH mean values, you should also include forested regions and regions wihout snow, and even oceans and sea ice, and also consider the impact of clouds. It is obvious that your reported NH values are larger than the actual NH mean forcing.

26. p. 34, lines 5–6. Referring to the previous comment, I would much prefer the

formulation "for the Northern Hemisphere ISCAs as a whole ...".

27. p. 35, lines 11–13. Climate models cannot incorporate remote sensing retrievals directly. They could however be used for model validation and to guide model development.

28. p. 62, caption of Fig. 7. It should be indicated whether the lower panel refers to instantaneous or daily radiative forcing.

29. p. 63, Fig. 8: A couple of things to be checked: 1) Are the Flanner et al. results all-sky or clear-sky values; 2) do the $RF_{\mathrm{MODIS,daily}}$ values represent January-February (as in the rest of the paper) or spring? It would not be meaningful to compare Jan-Feb vs. March-May.

30. Fig. S5: It is inconsistent to compare springtime radiative forcing in (a) with radiative forcing based on CMIP6 (i.e., CESM2) soot content in snow in January-February in (b).

**Technical and language corrections**

1. p. 4, lines 2-3: This sentence is cumbersome. Suggestion: "As a result, persistent uncertainties remain in regional and global-scale RFLS estimates based on field measurements."

2. p. 4, line 7: add "explaining" before "approximately one quarter of observed global warming".

3. p. 6, line 18: replace "valuable parameters" with "valuable information". (The reason

is explained in the specific comment #27).

4. p. 7, line 18: replace "generated by" with "derived from".

5. p. 8, line 4: replace "solar radiation" with "solar radiances".

6. p. 11, lines 11-12: "standard aerosol types"?

7. p. 12, line 4: "indicent radiation, surface spectral distribution". Do you mean "incident radiation at the surface and its spectral distribution"?

8. p. 14, line 5: I think this should be "both of which are required to exceed 0.6".

9. p. 18, lines 4-6: I think this sentence should be moved after Eq. (15): "The spatial variability in snow albedo due to $I_{\mathrm{LAPs}}$ can be expressed as

$$Eq.(15)$$

where $\overline{R_{\mathrm{eff}}}$, $\overline{SD}$ and $\overline{G}$ indicate spatial-mean values of $R_{\mathrm{eff}}$, $SD$, and $G$, with $\overline{G}$ requiring spatially constant values for the solar zenith angle, surface topography, and solar radiation parameters".

10. p. 22, line 5: "respectably" should be "respectively".

11. p. 35, line 3: It is not clear what "synthetically" means here.

12. Figure 1 (and also Fig. S1) would be easier to read if the values given on the colour bars would match with the values used to draw the curves. Now it is difficult to say which LAP content, snow depth etc. each curve exactly represents. Also, in the caption of Fig. 1, "angel" should be "angle".

[Figure]

13. In Fig. 3, "savannas" should probably be "tundra"?

14. In Fig. 6 and also Fig. S5, the interpretation of the box-plots should be explained.

15. In Fig. S3b, "confindence level" should be "confidence level".

---

## Referee Comment (RC2) · Edward Bair (Referee) · 20 May 2020

Aside from the Data Statement section, the authors responded to my comments, but did not make changes to the manuscript addressing my suggestions in my Access Review. Given the preliminary nature of the Access Review, that's fine with me, but during this formal review stage I ask that my comments be addressed in the manuscript. I have provided a copy of the Access Review below.

Limitations:

In the manuscript, the authors have not addressed the problem of distinguishing between absorbers in the air and on the snowpack as stated by Warren (2013). Some of the regions examined have extensive air pollution. The MCD43C3 albedo product used relies on MOD09 surface reflectance which masks out snow when estimating aerosol optical thickness. Snow is difficult to mask, and the "dark and dense vegetation technique" used to estimate aerosol optical thickness (Vermote & Saleous, 2006) has shown errors over snow cover in the past that have supposedly been addressed (Vermote et al., 2002). However, one can still find errors. For example, MOD09 surface spectra sometimes show strong hook features over relatively clean fully-snow covered pixels in the visible wavelengths that are not present in the top of atmosphere reflectances and can only be ascribed to problems with atmospheric correction. I'm not suggesting that the MCD43C3 product is unsuitable, rather I'd like to see its limitations over snow discussed in the manuscript. Saying that "more [sic] in-situ observations and hyperspectral imagery are needed.." is not a sufficient response.

MYD10C1 does not use spectral unmixing; it uses the 2-band NDSI which shows high scatter when converted to fractional snow cover using Equation 5. Thus, the fractional snow cover filter used is an undiscussed bias in the approach, where LAP could be mistaken for non-snow objects and vice-versa.

Note that MODIS is a multispectral, not a hyperspectral sensor.

Exclusion of midlatitude mountains and other vast snow-covered areas in the Northern Hemisphere is substantial and should be stated in the Abstract. As Referee #1 and #2 both point out, the domain (non-vegetated & non-mountainous areas) and time periods (Jan-Feb) are limited. These limitations undermine global application (e.g. p3 l1 & p6 l2).

NB 5/20/2020

_______________________ Access Review from 4/4/2020

Further discuss limitations

[Figure]

Consider addressing the challenges in measuring LAP stated in Warren (2013) directly, such as distinguishing between absorbers in the air and those in the snowpack.

Equation 5 when applied in the MOD10A1 product shows an RMSE of 0.227 and a positive bias of 0.11 (Rittger et al., 2013). These errors and biases are important because darker objects at visible wavelength in mixed snow-covered pixels (e.g. shadows and vegetation) can be misidentified as LAP.

Section 3.2.3

This basis for the snow grain size retrievals cites studies (Nolin & Dozier, 2000; Painter et al., 2013; Seidel et al., 2016) which use hyperspectral imagery at an more than order magnitude greater spatial and spectral resolution than a multispectral instrument like MODIS. The authors are relying on the albedo retrieval from a single MODIS band at 1.24 $\mu$m to estimate grain size. This approach has high uncertainty due to errors in albedo retrievals from MODIS. In a previous study, Pu et al. (2019) state the MAE is 71 $\mu$m or 3 times greater than in the studies cited above using hyperspectral instruments.

The previous two comments suggest why substantial correction factors for the remotely-sensed measurements (Section 4.3) are needed.

Section 4.1

The study area does not include most of the midlatitude mountains in the northern hemisphere. Snow and ice melt in these areas provides a valuable water resource to over 1B people worldwide (Barnett et al., 2005) and studies cited by the authors in the Introduction (Painter et al., 2012; Seidel et al., 2016) show this snow is heavily affected by LAP.

Data availability

No data statement is provided. Please see the ACP Data Policy which requires a statement of how the data can be accessed.

[Figure]

Barnett, T. P., Adam, J. C., & Lettenmaier, D. P. (2005). Potential impacts of a warming climate on water availability in snow-dominated regions. Nature, 438, 303-309. https://doi.org/10.1038/nature04141

Nolin, A. W., & Dozier, J. (2000). A Hyperspectral Method for Remotely Sensing the Grain Size of Snow. Remote Sensing of Environment, 74, 207-216. https://doi.org/https://doi.org/10.1016/S0034-4257(00)00111-5

Painter, T. H., Bryant, A. C., & Skiles, S. M. (2012). Radiative forcing by light absorbing impurities in snow from MODIS surface reflectance data. Geophysical Research Letters, 39, L17502. https://doi.org/10.1029/2012GL052457

Painter, T. H., Seidel, F. C., Bryant, A. C., McKenzie Skiles, S., & Rittger, K. (2013). Imaging spectroscopy of albedo and radiative forcing by light-absorbing impurities in mountain snow. Journal of Geophysical Research: Atmospheres, 118, 9511-9523. https://doi.org/10.1002/jgrd.50520

Pu, W., Cui, J., Shi, T., Zhang, X., He, C., & Wang, X. (2019). The remote sensing of radiative forcing by light-absorbing particles (LAPs) in seasonal snow over northeastern China. Atmos. Chem. Phys., 19, 9949-9968. https://doi.org/10.5194/acp-19-9949-2019

Rittger, K., Painter, T. H., & Dozier, J. (2013). Assessment of methods for mapping snow cover from MODIS. Advances in Water Resources, 51, 367-380. https://doi.org/10.1016/j.advwatres.2012.03.002

Seidel, F. C., Rittger, K., Skiles, S. M., Molotch, N. P., & Painter, T. H. (2016). Case study of spatial and temporal variability of snow cover, grain size, albedo and radiative forcing in the Sierra Nevada and Rocky Mountain snowpack derived from imaging spectroscopy. The Cryosphere, 10, 1229-1244. https://doi.org/10.5194/tc-10-1229-2016

Vermote, E. F., El Saleous, N. Z., & Justice, C. O. (2002). Atmospheric correction of MODIS data in the visible to middle infrared: first results. Remote Sensing of Environment, 83, 97-111. https://doi.org/https://doi.org/10.1016/S0034-4257(02)00089-5

Vermote, E. F., & Saleous, N. (2006). Operational Atmospheric Correction of MODIS Visible to Middle Infrared Land Surface Data in the Case of an Infinite Lambertian Target. In J. J. Qu, W. Gao, M.

Kafatos, R. E. Murphy, & V. V. Salomonson (Eds.), Earth Science Satellite Remote Sensing: Vol. 1: Science and Instruments (pp. 123-153). Berlin, Heidelberg: Springer Berlin Heidelberg. https://doi.org/10.1007/978-3-540-37293-6_8

Warren, S. G. (2013). Can black carbon in snow be detected by remote sensing? Journal of Geophysical Research: Atmospheres, 118, 779-786. https://doi.org/10.1029/2012jd018476
* * *

---

## Author Response (AR1)

**Editor: Nikos Hatzianastassiou**

Thank the Editor very much for handling the manuscript We take into account all the comments from referees and make revisions. Please check the responses to the referees and the revised manuscript.

**Anonymous Referee #1**

We are very grateful for the referee's critical comments. The followings are our point-by-point responses to the comments. Our responses start with "R:".

**General comments**

This paper presents a method for estimating the radiative forcing due to light-absorbing particles (LAPs) in snow (RFLS) using several data sources, which include MODIS albedos, snow grain size derived from MODIS data, snow depth from the ERA-Interim reanalysis, surface downwelling solar radiation from CERES, and finally, in situ measurements of BC in snow (used for computing correction factors for the algorithm). The proposed approach allows the estimation of RFLS in larger areas than would be possible with in situ measurements alone. It thus provides an additional data source complementing estimates from in situ data and climate models. As noted in the introduction, there are previous studies that utilized MODIS to retrieve the radiative forcing of LAPs in snow, but this might be the first one to consider the spatial variability in RFLS between different regions. The approach is further employed to analyze the factors underlying the spatial variation of RFLS, finding that the variations in LAP content, snow depth and geographical factors (e.g., latitude) are more important than those in snow grain size (Fig. 7). Furthermore the retrieved values of RFLS are compared with results from a few climate models (Figs. 8 and 9) and with previous studies (Fig. 10).

A practical limitation of the proposed approach is that it can only be applied in regions with no/very short vegetation. Also, judging by the correction factors needed to eliminate systematic differences to RFLS derived from in situ data, it appears that the approach works fairly well in heavily polluted regions, but for regions with relatively clean snow, the uncertainties are very large (Fig. S2b). So if one interpretes

"hemispherescale" values (p. 6, line 17) as "hemisphere-mean" values, they cannot yet be obtained with this approach.

There is certainly enough new material in this work to be published in ACP. The paper is reasonable well written especially as regards the description of the approach, but I think there are disturbingly many numerical values in the text towards the end, and possibly some apples-to-oranges comparisons.

R: Thank you very much for the positive comments, which will encourage us to do more in-depth research in the future. Moreover, the referee's comments are quite significant that can help us to improve the paper quality substantially. We have addressed all of the comments carefully according to the suggestions. Especially, we have extended the study period from January-February to December-May, so that the snow cover area over the Arctic can be retrieved. We have replaced the clear-sky radiative forcing with all-sky radiative forcing, which makes more sense to the research community. We have recalculated the broadband snow albedo with wavelengths of 300-2500 nm. We have revised misleading descriptions and reduce some numerical values throughout the manuscript according to the suggestions. All of the detailed responses can be seen as follow.

1. It is not justified to "sell" the values averaged over all ISCAs as the Northern Hemisphere (NH) mean values (e.g., p.2, lines 11-13, and p. 34, lines 6-7) since they really represent only a small part of the NH land area. The approach samples only areas with (nearly) full snow cover and no/very short vegetation, which naturally results in a high bias in the computed "NH average" RF. The assumption of clear-sky conditions further increases the RF values, while the analysis of only January and February data decreases the RF in the Arctic, but perhaps increases it at midlatitudes, compared to annual-mean values. In general, you should avoid listing numerical values without explaining what they really mean, especially in the abstract.

R: The referee's opinions are very valuable. We have replaced "Northern Hemisphere (NH) averaged radiative forcing" with "radiative forcing averaged over mapped snow-covered area in Northern Hemisphere" and revised the similar issues throughout the manuscript. Moreover, we have recalculated the all-sky radiative forcing to replace the clear-sky radiative forcing and extended the study period of only January and February to December to May. In addition, we have revised the abstract and main text carefully to avoid the values without certain explanation.

2. Specifically, the abstract should state that these are clear-sky values, that the albedo reduction refers to wavelengths 300-1300 nm, and the RF values refer to areas with full snow cover and little/no vegetation above snow.

R: We have recalculated the broadband snow albedo with wavelengths of 300-2500 nm under all-sky condition. We have stated the "estimated radiative forcing" as "…radiative forcing except for midlatitude mountains in December-May for the period 2003–2018…over mapped snow-covered area in Northern Hemisphere " in the abstract and throughout the manuscript.

3. p. 6, lines 14-16, Section 2.4, and Section 4.6: Taking only two CMIP6 models, and calling them "CMIP6" or "CMIP6 ensemble mean" is misleading, especially as the two models (CESM2 and CESM2-WACCM) are very closely related and produce nearly identical results (Fig. S4). It would be advantageous to use data from more CMIP6 models, if data from more models has now become available. If not, just take CESM2 and call it CESM2! In addition, instead of a "Global climate model", you should use the specific model name for Flanner et al. (2009), that is CAM3.1. Incidentally, it is a predecessor of the atmospheric and land components of CESM2.

R: Thanks very much for the explanations and suggestions. Actually, we have limited knowledge of climate models and the referee's comments help us improve the

understanding about CESM2 and CAM3.1. We have removed the comparison about CAM3.1 because it is the predecessor of the atmospheric and land module of CESM2 as the referee mentioned. We have carefully revised the improper description throughout the manuscript.

4. p. 7, line 6 and elsewhere: Why do you only use data for January and February? The reason for this should be stated explicitly. Perhaps because the midlatitude snow cover is most extensive then? However, this choice screens out almost all of the Arctic, due to the low sun angles, so that the "Arctic" RFLS values is this work in practice only represent southern Greenland. Also, considering spring months would increase the Arctic RFLS values substantially

R: The referee's comments are quite significant. We have updated the data from January-February to December-May, so that the study period can include winter and spring, and snow-covered areas over the Arctic have been mapped.

5. p. 9. A brief description of the in situ BC measurements employed to correct the RFLS values should be included in Section 2 (at least, regions and references), perhaps between current Sections 2.2 and 2.3. Do these measurements represent BC or LAPs in general?

R: We have added more details about in-situ measurements in Sect. 2.3. These measurements are equivalent BC, which can represent the all light absorption by LAPs. We have added a detailed explanation for "equivalent BC" in p. 9, lines 17-21.

6. p. 9, lines 7–10: What was the reason for converting SWE to snow depth? To my knowledge, this has no effect on the results (in the end, SNICAR cares of SWE only).

R: Actually, SNICAR cares of SWE only. However, the offline SNICAR requires both snow depth and snow density as input, so that we converted SWE to snow depth with an assumed snow density. Anyhow, as your say, this has no effect on the results.

7. p. 9, Section 2.3. It should be stated how/why these emission data were used. I get the impression that they were used just as background information (not in estimating the RFLS).

R: Indeed, BC emission and deposition data were used just as background information. So that we have moved Figure 4a and 4b to the supplements (Figure S2a, b).

8. p. 11, lines 11-16: You describe how SBDART has several options for defining the atmospheric properties. It would be more important to tell what was assumed in the present calculations (also regarding aerosols).

R: We have added the description about the options for defining the atmospheric properties in SBDART. Details can be seen in p. 11, lines 3-7:

"In our study, the subarctic and midlatitude winter standard atmospheric condition is performed as well as the tropospheric and stratospheric background aerosols are archived in SBDART (Tanre, D. et al., 1990). According to Dang et al. (2017), the cloud optical depth in high-latitude and mid-latitude was assumed as 11 and 20 under cloudy-sky condition, respectively."

9. p. 12, line 11: You could add snow grain shape to this list.

R: We note that SNICAR only assumes a spherical snow grain. We have added the description about snow grain shape as "…and spherical grain shape.'' in p. 11, line 20.

10. p. 12, lines 20-21: "previous studies have tended to assume a semi-infinite snowpack". This is a good point, and I think it would be worth showing how much this influences the results. Consider adding a figure which shows the ratio of RFLS computed using the actual (ERA-Interim) snow depth vs. RFLS computed using semi-infinite snow.

R: As the referee's suggestion, we have added Figure 7 to show the ratio of RFLS computed using the actual (ERA-Interim) snow depth vs. RFLS computed using semi-infinite snow and taken a discussion about the influence of snow depth on radiative forcing retrieval in Sect. 4.4.

11. p. 13, line 20: add "...for clear-sky conditions" at the end of the sentence.

R: We have replaced clear-sky radiative forcing with all-sky radiative forcing throughout the manuscript.

12. p. 16, Eq. (7): Please state explicitly that the impact of LAPs on snow albedo computed in this work refers to the spectral range 300-1300 nm only. There is a chance of misinterpretation here, as usually people think of broadband albedo integrated over the entire downwelling solar spectrum at the surface. (An alternative would be to calculate "real" broadband albedo changes, integrated over 0.3–4 µm or at least 0.3–2.5 µm). This choice should not matter for RFLS, however.

R: Thanks for the referee's suggestion. We have recalculated the broadband albedo with wavelengths of 300-2500 nm.

13. In Eq. (2), diffuse and direct spectral solar radiation are added as such ($E_{dif}(\lambda; \varphi)+E_{dir}(\lambda; \varphi)$), suggesting that they both are defined wrt. a horizontal surface, but in Eq. (7) (and Eqs. (10) and (11)) the direct radiation is weighted by the cosine of local solar zenith angle ($E_{dir;\lambda} \cos \beta + E_{dif;\lambda}$), which implies that the direct radiation is defined wrt. a surface perpendicular to Sun's direction. This seems inconsistent.

R: We have revised this inconsistency throughout the manuscript.

14. p. 17, line 7: "we assumed that the properties for snow and LAPs remain invariable throughout the day". In fact, if you keep the snow physical properties and LAP concentration constant, the impact of LAPs on snow albedo decreases with increasing solar zenith angle, so the use of $\Delta\alpha_{MODIS,corrected}^{LAPs}$ evaluated at noon probably overestimates the daily-average impact of LAPs somewhat. (I would guess, perhaps of the order of 10%, but this is something that you could check with SNICAR.)

R: We have corrected the overestimates by further simulating the daily-average snow albedo by changing the solar zenith angle from sunrise to sunset using SNICAR model and SBDART model. Revisions are added in p. 16, lines 11-16 and as follow:

"Following Miller et al. (2016), we assumed that the properties for snow and LAPs remain invariable throughout the day. Based on calculated $\alpha_{snow,\lambda}^{mdl}$ and $\alpha_{snow,\lambda}^{MODIS}$ at noon, the diurnal variation of pure and polluted snow albedo can be simulated by SNICAR from sunrise to sunset. Then, daily-average snow albedo reduction ($\Delta\alpha_{MODIS,daily}^{LAPs}$) can be derived by integrating the diurnal snow albedo reduction, which is weighted by simultaneous solar irradiance from SBDART."

15. In Eq. (11), is cos β the daytime mean value?

R: We have new algorithm. cos β is calculated based on the certain latitude and solar zenith (solar azimuth) from sunrise to sunset.

16. p. 22, lines 16-17: You should remind the reader that this result refers specifically to the months of January and February. In spring and early summer, much of the Arctic is still snow-covered and solar radiation is much more abundant, so RFLS is substantially larger than in January-February.

R: We have updated the data from January-February to December-May, so that the study period can include winter and spring, and snow-covered areas over the Arctic have been mapped.

17. p. 22, line 20. "In situ observations of snow albedo reduction" actually refer to the albedo reduction calculated using in-situ observed LAPs. Here, it should be noted what was the measure of LAPs used in the in situ observations? Was it BC (excluding dust) or equivalent BC (implicitly also including dust). I guess in-situ obervations usually yield the latter?

R: We have revised "in situ observations of snow albedo reduction" as "Albedo reduction calculated using in-situ observed LAPs ($\Delta \alpha_{in-situ,daily}^{LAPs}$)..." in p. 22, line 15. Also, we have added a statement that the measure of LAPs was equivalent BC in p. 9, lines 17-21.

18. p. 22, line 11. These corrections deserve a bit more discussion. The value $c_{polluted}$ = 1:1 suggests that the approach works rather well for heavily polluted snow. However, the value $c_{clean}$ = 5:6 for "relatively pure" snow, along with the scatter of points in Fig.

S2, suggests that the method becomes quite inaccurate then. Can you comment on the possible reasons for that? Perhaps the limiting factor is simply the accuracy of albedo calculations and observations, and a possible systematic bias between the two? For example, for 100 ng/g of BC (which many would already consider not so clean snow!) the albedo reduction is only ∼0.02. So, if in Eq. (7) $\alpha_{snow;\lambda}$ mdl is biased high and/or $\alpha_{snow;\lambda}$ MODIS is biased low, this would result in c > 1, the more so the cleaner the snow.

R: We have added a discussion about the uncertainty of the snow albedo reduction retrieval, which is negative correlated to snow pollution condition, to demonstrate the low correction value for heavily polluted snow but high correction value for relatively pure snow. We also discussed the influence of in-situ observation on the correction factor as suggested. Details can be seen in p. 27, lines 18-21 and p. 28, lines 1-18.

19. p. 23–26: I think the large number of numerical values in the text is disrupting to the reader. Some concrete suggestions would be: 1) for p. 23, lines 14-20 provide the MAE and RMS statistics in the figure panels in Fig. 5, 2) in Section 4.4., put the numerical values in a table. If you prefer to keep them in the text, you could at least skip the instantaneous RF values.

R: We have simplified the number of numerical values in the text for avoiding to disrupt the reader throughout the manuscript and put the MAE and RMSE statistics in Table S1 in supplements as suggestions. We have put the general statistics of snow albedo reduction and radiative forcing in Sect. 4.4 in Table 1 and we prefer to keep the values in different regions in the text in detail. Finally, we removed the discussion of instantaneous RF values in the text.

20. p. 28, line 10 – p. 29, line 7: As noted above, the model used by Flanner et al. (2009) should be called "CAM3.1" rather than "GCM". More importantly, you discuss

springtime RF for Flanner et al. Did you compute springtime values for the MODIS retrievals too? This should be made clear in the text. Comparing January–February values with springtime (March–May?) values would be meaningless.

R: We have removed the comparison about CAM3.1 because it is the predecessor of the atmospheric and land module of CESM2 as the referee mentioned.

21. p. 31-32 and Fig. 10. The comparison with previous radiative forcing estimates is interesting, but one should be careful not to compare apples with oranges let alone watermelons – or at least be explicit about when this is being done. In other words, I think you should provide more information about the previous studies considered here. The RF differences could arise from the consideration of different regions, different seasons, clear-sky vs. all-sky forcing etc., so these details should be mentioned. This information would probably best fit in a table.

R: As the referee's suggestions, we have added a table (Table 2) about the detailed information of the previous studies and revised the discussion about the possible sources of the RF differences. Details can be seen in Sect. 5.

22. p. 31, line 21 – p. 32, line 1: "Miller et al. (2016) reported a daily RFLS of < 4

W m−2". Figure 10b (2nd panel) shows much larger values.

R: We have replaced Figure 10 with Table 2. We have rechecked the reference and revised as follow:

"…Miller et al. (2016) reported a daily RFLS of ~35-86 (37-100) W m$^{-2}$ based on in-situ measurements (remote sensing) in the San Juan Mountains in May 2010."

23. p. 32, line 7: Should this be Qian et al. (2014) or Qian et al. (2009) (cf. Fig 10c, second panel).

R: Thank you for pointing out the mistake, it should be "Qian et al. (2009)". We have rechecked all data in section 5.

24. p. 32, lines 16-17: It is stated that Wang et al. (2014a) reported a northern hemisphere RFLS value of 0.45 W m$^{-2}$. However, so far I can tell, that paper is concerned with the direct radiative forcing due to BC in air (not snow). Furthermore, Fig. 10c refers to Wang et al. (2004), which is not present in the reference list.

R: Thank you for pointing out the mistake. "Wang et al. (2004)" should be "Wang et al. (2014a)". In addition, Wang et al. (2014a) only reported the RF due to BC in air actually and has nothing to do with snow. We have removed it and rechecked all data in section 5.

25. p. 32, lines 15–21. I think your explanation is in principle correct, although at least the values of Bond et al. (2013) and Hansen and Nazarenko (2004) are annual-mean values, not January-February. But the fundamental point here is that your approach cannot provide northern-hemisphere (NH) mean values, which the cited studies attempt to provide, uncertainties notwithstanding. It can only provide values for ISCAs that are snow-covered and without much vegetation. For true NH mean values, you should also include forested regions and regions without snow, and even oceans and sea ice, and also consider the impact of clouds. It is obvious that your reported NH values are larger than the actual NH mean forcing.

R: We have revised the description of our RF as "radiative forcing averaged over mapped snow-covered area in Northern Hemisphere" and added a table about the detailed information of the RF from previous studies to demonstrate the difference.

26. p. 34, lines 5–6. Referring to the previous comment, I would much prefer the formulation "for the Northern Hemisphere ISCAs as a whole ...".

R: According to your suggestion, we have revised "For the Northern Hemisphere as a whole …" as "For the Northern Hemisphere ISCAs as a whole ...".

27. p. 35, lines 11–13. Climate models cannot incorporate remote sensing retrievals directly. They could however be used for model validation and to guide model development.

R: This sentence has been revised as "We propose that climate models validated by these refined remote sensing retrievals should be able to capture the RFLS more accurately, thereby providing more reliable estimates of the future impacts of global climate change."

28. p. 62, caption of Fig. 7. It should be indicated whether the lower panel refers to instantaneous or daily radiative forcing.

R: We have revised Figure 7. The attribution refers to daily RF.

29. p. 63, Fig. 8: A couple of things to be checked: 1) Are the Flanner et al. results allsky or clear-sky values; 2) do the $RF_{MODIS,daily}$ values represent January-February (as in the rest of the paper) or spring? It would not be meaningful to compare Jan-Feb vs. March-May.

R: We have removed the comparison about CAM3.1 because it is the predecessor of the atmospheric and land module of CESM2 as the referee pointed out.

30. Fig. S5: It is inconsistent to compare springtime radiative forcing in (a) with radiative forcing based on CMIP6 (i.e., CESM2) soot content in snow in January-February in (b).

R: We have revised Figure S5. We have removed the comparison about CAM3.1 because it is the predecessor of the atmospheric and land module of CESM2 as the referee pointed out. When comparing with CESM2, the MODIS retrievals are the averages of December-May.

1. p. 4, lines 2-3: This sentence is cumbersome. Suggestion: "As a result, persistent uncertainties remain in regional and global-scale RFLS estimates based on field measurements."

R: We have revised this sentence as suggestion.

2. p. 4, line 7: add "explaining" before "approximately one quarter of observed global warming".

R: Added as suggestion.

3. p. 6, line 18: replace "valuable parameters" with "valuable information". (The reason is explained in the specific comment #27).

R: Revised as suggestion.

4. p. 7, line 18: replace "generated by" with "derived from".3

R: Revised as suggestion.

5. p. 8, line 4: replace "solar radiation" with "solar radiances".

R: Revised as suggestion.

6. p. 11, lines 11-12: "standard aerosol types"?

R: Thank you pointed out the grammatically wrong sentence. We have Revised.

7. p. 12, line 4: "indicent radiation, surface spectral distribution". Do you mean "incident radiation at the surface and its spectral distribution"?

R: Revised as suggestion.

8. p. 14, line 5: I think this should be "both of which are required to exceed 0.6".

R: Revised as suggestion.

9. p. 18, lines 4-6: I think this sentence should be moved after Eq. (15): "The spatial variability in snow albedo due to ILAPs can be expressed as Eq:(15) where $R_{eff}$, SD and G indicate spatial-mean values of $R_{eff}$, SD, and G, with G requiring spatially constant values for the solar zenith angle, surface topography, and solar radiation parameters".

R: Revised as suggestion.

10. p. 22, line 5: "respectably" should be "respectively".

R: Revised.

11. p. 35, line 3: It is not clear what "synthetically" means here.

R: We want to express "relatively comprehensive and systematically". If the referee still consider "synthetically" in unreadable we will revise it in next version.

12. Figure 1 (and also Fig. S1) would be easier to read if the values given on the colour bars would match with the values used to draw the curves. Now it is difficult to say which LAP content, snow depth etc. each curve exactly represents. Also, in the caption of Fig. 1, "angel" should be "angle".

R: Revised as suggestion.

13. In Fig. 3, "savannas" should probably be "tundra"?

R: Revised and Figure 3b has been replotted.

14. In Fig. 6 and also Fig. S5, the interpretation of the box-plots should be explained.

R: The interpretation of the box-plots had been added in this version.

15. In Fig. S3b, "confindence level" should be "confidence level".

R: Revised.

**Edward Bair (Referee)**

nbair@eri.ucsb.edu

We are very grateful for the referee's critical comments. The followings are our point-by-point responses to the comments. Our responses start with "R:".

Aside from the Data Statement section, the authors responded to my comments, but did not make changes to the manuscript addressing my suggestions in my Access Review. Given the preliminary nature of the Access Review, that's fine with me, but during this formal review stage I ask that my comments be addressed in the manuscript. I have provided a copy of the Access Review below.

R: We are really sorry for making no changes to the manuscript addressing your suggestions aside from the Data Statement section. We really appreciate the reviewer's comments, which can help us to improve the paper quality substantially and encourage us to do more in-depth research in the future. We have addressed all the comments very carefully in this section as detailed below.

Limitations:

In the manuscript, the authors have not addressed the problem of distinguishing between absorbers in the air and on the snowpack as stated by Warren (2013). Some of the regions examined have extensive air pollution. The MCD43C3 albedo product used relies on MOD09 surface reflectance which masks out snow when estimating aerosol optical thickness. Snow is difficult to mask, and the "dark and dense vegetation technique" used to estimate aerosol optical thickness (Vermote & Saleous, 2006) has shown errors over snow cover in the past that have supposedly been addressed (Vermote et al., 2002). However, one can still find errors. For example, MOD09 surface spectra sometimes show strong hook features over relatively clean fully-snow covered pixels in the visible wavelengths that are not present in the top of atmosphere

reflectances and can only be ascribed to problems with atmospheric correction. I'm not suggesting that the MCD43C3 product is unsuitable, rather I'd like to see its limitations over snow discussed in the manuscript. Saying that "more [sic] in-situ observations and hyperspectral imagery are needed." is not a sufficient response.

R: The referee's opinions are very valuable. Indeed, the absorbers in the air can disturb the retrieval of MODSI surface reflectance. According to the MODIS Surface Reflectance User's Guide (Collection 6, https://modis.gsfc.nasa.gov/data/dataprod/mod09.php), the accuracy of the atmospheric correction is typically: $\pm(0.005 + 0.05*\text{reflectance})$ under conditions that AOD is less than 5.0 and solar zenith angle is less than 75°. Therefore, we estimate the uncertainty of calculated radiative forcing based on the level of accuracy of the atmospheric correction in our study. Details could be found in Section 4.5.

MYD10C1 does not use spectral unmixing; it uses the 2-band NDSI which shows high scatter when converted to fractional snow cover using Equation 5. Thus, the fractional snow cover filter used is an undiscussed bias in the approach, where LAP could be mistaken for non-snow objects and vice-versa.

R: The referee's opinions are very valuable. We have added an estimation and discussion on the uncertainty of calculated radiative forcing from the uncertainty of converting NDSI to fractional snow cover. According to (Rittger et al., 2013) and Riggs et al. (2016), the converted percentage error assumed in this study was 10%. Details could be found in Section 4.5.

Note that MODIS is a multispectral, not a hyperspectral sensor.

R: We have revised the mistake.

Exclusion of midlatitude mountains and other vast snow-covered areas in the Northern Hemisphere is substantial and should be stated in the Abstract. As Referee #1 and #2 both point out, the domain (non-vegetated & non-mountainous areas) and time periods

(Jan-Feb) are limited. These limitations undermine global application (e.g. p3 l1 & p6 l2).

R: We have added a statement for exclusion of midlatitude mountains in the Abstract and an explanation for why we exclude midlatitude mountains in the main text in p. 20, lines 7-11. We have extended the study period from January-February to December-May, so that the snow cover area over the Arctic can be retrieved. Also, we have replaced the clear-sky radiative forcing with all-sky radiative forcing, which makes more sense to the research community.

Further discuss limitations

Consider addressing the challenges in measuring LAP stated in Warren (2013) directly, such as distinguishing between absorbers in the air and those in the snowpack. Equation 5 when applied in the MOD10A1 product shows an RMSE of 0.227 and a positive bias of 0.11 (Rittger et al., 2013). These errors and biases are important because darker objects at visible wavelength in mixed snow-covered pixels (e.g. shadows and vegetation) can be misidentified as LAP.

R: As mentioned above, we have added estimations and discussions about the uncertainty of calculated radiative forcing from converting NDSI to fractional snow cover in Section 4.5.

Section 3.2.3

This basis for the snow grain size retrievals cites studies (Nolin & Dozier, 2000; Painter et al., 2013; Seidel et al., 2016) which use hyperspectral imagery at an more than order magnitude greater spatial and spectral resolution than a multispectral instrument like MODIS. The authors are relying on the albedo retrieval from a single MODIS band at 1.24 $\mu$m to estimate grain size. This approach has high uncertainty due to errors in albedo retrievals from MODIS. In a previous study, Pu et al. (2019) state the MAE is 71$\mu$m or 3 times greater than in the studies cited above using hyperspectral instruments. The previous two comments suggest why substantial correction factors for the remotely-sensed measurements (Section 4.3) are needed.

R: The referee's opinions are very valuable. We have added an estimation and discussion about the uncertainty of calculated radiative forcing from snow grain size retrieval. The percentage error of snow grain size retrieval assumed in this study is 30% according to the study of Wang et al. (2017) and Pu et al. (2019). Based on the discussion of the uncertainties from atmospheric correction, snow cover fraction calculation and snow grain size retrieval, we further demonstrate the necessity of substantial correction factors for the remotely-sensed measurements and why the correction factor is different over relatively polluted snow and relatively clean snow. Details could be found in Section 4.5.

Section 4.1

The study area does not include most of the midlatitude mountains in the northern hemisphere. Snow and ice melt in these areas provides a valuable water resource to over 1B people worldwide (Barnett et al., 2005) and studies cited by the authors in the Introduction (Painter et al., 2012; Seidel et al., 2016) show this snow is heavily affected by LAP.

R: In this study, the MODIS surface albedo data used is MCD43C3, which has a resolution of $0.05° \times 0.05°$. Usually, the snow cover faction over midlatitude mountains at such a coarse resolution is low, which cause that most of midlatitude mountains are not mapped as snow-covered area. In addition, midlatitude mountains are characterized as complex terrain, which will cause high biases in radiative forcing retrieval at a coarse resolution of $0.05° \times 0.05°$ in spite of topographic correction. Therefore, we didn't report the results over midlatitude mountains in this study. We have added an explanation why midlatitude mountains are not included in Section 4.1 in p. 20, lines 7-11. However, we agreed with the referee that the radiative forcing over midlatitude mountains are quite important, so that we will focus on these areas using finer resolution MODIS data (MCD43A3, MOD/MYD09) or data from high resolution satellites (e.g. Sentinel-2 and Landsat 8) in the future.

Data availability

No data statement is provided. Please see the ACP Data Policy which requires a statement of how the data can be accessed.

R: We have added more descriptions about the data access referring to the ACP Data Policy in Data availability.

References:

E. Vermote. (2015). MOD09A1 MODIS Surface Reflectance 8-Day L3 Global 500m SIN Grid V006. NASA EOSDIS Land Processes DAAC. http://doi.org/10.5067/MODIS/MOD09A1.006

[revised manuscript text omitted]
_{\sim snow,\lambda}^{\sim all}\,\alpha_{MODIS,\lambda}^{all}\,\alpha_{\sim blue-sky,\lambda}$$

$$= \frac{E_{all-sky,\lambda}\cancel{E_{\lambda}} \cdot FSC \cdot \alpha_{\sim snow,\lambda}^{\sim all}\alpha_{snowMODIS,\lambda}^{MODISall} + E_{all-sky,\lambda}\cancel{E_{\lambda}} \cdot (1 - FSC) \cdot \alpha_{underlying,\lambda}}{E_{all-sky,\lambda}\cancel{E_{\lambda}}}$$

$$= FSC \cdot \alpha_{\sim snow,\lambda}^{\sim all}\alpha_{MODISsnow,\lambda}^{MODISall} + (1 - FSC) \cdot \alpha_{underlying,\lambda}$$

$$\tag{5\underline{6}}$$

$$\alpha_{\sim snow,\lambda}^{\sim all}\alpha_{snow,\lambda}^{MODIS} = \frac{\alpha_{MODIS,\lambda}^{all}\cancel{\alpha_{blue-sky,\lambda}} - (1 - FSC)\cdot\alpha_{underlying,\lambda}}{FSC}$$

$$\underline{\quad}(6\underline{7})$$

where $E_{all-sky,\lambda}\cancel{E_{\lambda}}$ is total solar irradiance under all-sky condition, a linear combination of direct/diffuse competent of solar irradiance  under clear-sky and cloudy-sky using similar strategy via Eq. (1)-(4). $\alpha_{underlying,\lambda}$ represents the

albedo of the underlying surface and was obtained from Siegmund and Menz (2005).

As depicted in Fig. 3b, vegetation and bare soil are the main types of underlying surface

in the ISCA.

**3.2.3. Retrieval of snow grain size**

The snow optical-equivalent grain size ($R_{eff}$) is retrieved by fitting SNICAR-simulated

snow albedo to MODIS-derived snow albedo at 1240 nm (the central wavelength of

MODIS band 5), following the protocol of Nolin and Dozier (2000). This retrieval

method is not influenced by liquid water and water vapor and has been employed

widely in previous studies (e.g., Painter et al., 2013; Seidel et al, 2016). Both Nolin and

Dozier (2000) and Pu et al. (2019) reported that the retrieved $R_{eff}$ compares favorably

with ground-based measurements of snow grain size. In this study, we chose to exclude

the ISCA, where MODIS-derived snow albedo at 1240 nm is <0.3, to avoid

misrepresenting $R_{eff}$ (Tedesco et al., 2007).

**3.2.4. Retrieval of snow albedo reduction **and RFLS**

The  spectrally integrated reduction in snow albedo due to LAPs

($\Delta\alpha_{MODIS,\overline{ins}noon}^{LAPs}$) is estimated for local-noon and all-sky conditions, using solar

irradiance and the difference between MODIS-derived spectral snow albedo

($\alpha_{snow,\lambda}^{\overline{MODIS}all}$) and simulated pure snow albedo ($\alpha_{snow,\lambda}^{mdl}$). Because MODIS provides only

four VIS bands, we fitted snow albedo data obtained via MODIS to a continuous 300–

 2500 nm spectrum ($\alpha_{snow,\lambda}^{MODIS}$ with a 10 nm interval) following the method provided

by Pu et al. (2019). Thereafter, the broadband albedo reduction due to LAPs retrieved

from MODIS ($\Delta\alpha_{MODIS,noon}^{LAPs}$)  can be calculated as follows:

$$\Delta\alpha_{MODIS,noon}^{LAPs} = \frac{\sum_{\lambda=300nm}^{\lambda=2500nm}\left(\alpha_{snow,\lambda}^{mdl} - \alpha_{snow,\lambda}^{MODIS}\right) \cdot E_{all-sky,\lambda} \cdot \Delta\lambda}{\sum_{\lambda=300nm}^{\lambda=2500nm} E_{all-sky,\lambda} \cdot \Delta\lambda} \qquad (8)$$

where $\alpha_{snow,\lambda}^{mdl}$ is the pure snow albedo simulated by SNICAR using MODIS-derived

$R_{eff}$ and ERA-Interim snow depth data, $\alpha_{snow,\lambda}^{MODIS}$ is the continuous snow albedo

derived from MODIS retrievals, and $\Delta\lambda$ is 10 nm.

Following Miller et al. (2016), we assumed that the properties for snow and LAPs

remain invariable throughout the day. Based on calculated $\alpha_{snow,\lambda}^{mdl}$ and $\alpha_{snow,\lambda}^{MODIS}$ at

noon, the diurnal variation of pure and polluted snow albedo can be simulated by

SNICAR from sunrise to sunset. Then, daily-average snow albedo reduction

($\Delta\alpha_{MODIS,daily}^{LAPs}$) can be derived by integrating the diurnal snow albedo reduction, which

is weighted by simultaneous solar irradiance from SBDART. Similarly, we used

measurements of LAPs in contaminated snow to calculate the

 in-situ reduction in snow albedo ($\Delta\alpha_{in-situ,\underline{ins}\cancel{daily}}^{LAPs}$). To derive a correction factor for

MODIS retrievals, we applied a similar validation strategy to that of Zhu et al. (2017):

$$c = \frac{1}{n}\sum_{i=1}^{n}\left(\frac{\Delta\alpha_{MODIS,\underline{ins}\cancel{daily}}^{LAPs}}{\Delta\alpha_{in-situ,\cancel{daily}\underline{ins}}^{LAPs}}\right) \tag{9}$$

where $c$ is the correction factor for $\Delta\alpha_{MODIS,\cancel{daily}\underline{ins}}^{LAPs}$ and $n$ is the number of the

respective  in-situ measurements. Accordingly, the corrected albedo reduction

($\Delta\alpha_{MODIS,corrected}^{LAPs}$) is calculated as follows:

$$\Delta\alpha_{MODIS,corrected}^{LAPs} = \frac{1}{c} \cdot \Delta\alpha_{MODIS,\cancel{daily}\underline{ins}}^{LAPs} \tag{10}$$

The daily-average, spectrally integrated RFLS ($RF_{MODIS,\underline{ins}\cancel{daily}}^{LAPs}$) is

calculated for all-sky conditions as follows:

$$RF_{MODIS,daily}^{LAPs} = \Delta\alpha_{MODIS,corrected}^{LAPs} \cdot \underline{SW_{all-sky}} \tag{11}$$

$$\cancel{RF_{MODIS,ins}^{LAPs} = \Delta\alpha_{MODIS,corrected}^{LAPs} \cdot \sum_{\lambda=300\,nm}^{\lambda=1300\,nm} (E_{dir,\lambda} \cdot \cos\beta + E_{dif,\lambda}) \cdot \Delta\lambda}$$

$$\cancel{RF_{MODIS,daily}^{LAPs} = \Delta\alpha_{MODIS,corrected}^{LAPs} \cdot (SW_{dir} \cdot \cos\beta + SW_{dif})} \quad \cancel{(12)}$$

where $SW_{\cancel{dir}\underline{all-sky}}$  represent the average-daily total

downward shortwave fluxes,  obtained from CERES under all-sky

conditions.

**3.2.5. Attribution of spatial variability in snow albedo reductions and radiative forcing**

As demonstrated above, reductions in snow albedo and RFLS are dependent primarily

on LAP content, $R_{eff}$, snow depth ($SD$), solar zenith angle, surface topography, and

solar irradiance, the latter three of which can be categorized as the geographic

factor ($G$). We used an impurity index ($I_{LAPs}$) to represent the LAP content of the

snowpack (Di Mauro et al., 2015; Pu et al., 2019), following the equation:

$$I_{LAPs} = \frac{\ln(\alpha_{snow,band4}^{MODISall})}{\ln(\alpha_{snow,band5}^{MODISall})} \tag{12}$$

[revised manuscript text omitted]

1  for the Northern Hemisphere ISCA in January–February December-May during the period 2003–

2  2018. The boxes denote the 25th and 75th quantiles, and the horizontal lines represent the 50th

3  quantiles (medians), the averages are shown as red dots; the whiskers denote the 5th and 95th

4  quantiles.

1
[Figure]

[Figure]

[Figure]

Figure 7. The spatial distribution of the ratio of retrieved radiative forcing using semi-infinite snow
to radiative forcing using ERA-Interim snow depth.

[Figure]

3  Figure 8. The overall uncertainty of radiative forcing retrieval due to  atmospheric

4  correction, MODIS-derived snow grain size retrieval and snow cover fraction calculation.

[Figure]

Figure 79. Fractional contributions of LAPs, snow grain size ($R_{eff}$), geographic factor ($G$), and snow depth ($SD$) to the spatial variations of (a) snow albedo reduction and (b) daily radiative forcing.

[Figure]

RF$_{GCM}$ (W m$^{-2}$)

[Figure]

1 Figure 8. Spatial distributions of (a) springtime radiative forcing ($RF_{GCM}$) due to LAPs in snow,

2 derived from a GCM run by Flanner et al. (2007), and scatterplot of (b) $RF_{MODIS,daily}$ versus

3 $RF_{GCM}$.

[Figure]

[Figure]

Figure 10. (a) Spatial distributions of average-daily radiative forcing ($RF_{\text{CESM2}}$), based on the CESM2 soot content of snow in December-May for the period 2003–2014. (b) Scatterplot of $RF^{LAPs}_{MODIS,daily}$ versus $RF_{\text{CESM2}}$.

1 Table 1. Statistics for regionally averaged (5th and 95th quantiles) albedo reduction

2 ($\Delta\alpha_{MODIS,corrected}^{LAPs}$) and daily radiative forcing ($RF_{MODIS,daily}^{LAPs}$, W m$^{-2}$)

|  | Northeastern China | EUA | NA |
|---|---|---|---|
| Albedo reduction ($\Delta\alpha_{MODIS,corrected}^{LAPs}$) | 0.11 (0.077~0.14) | 0.031 (0.017~0.049) | 0.027 (0.014~0. |
| Daily radiative forcing ($RF_{MODIS,daily}^{LAPs}$, W m$^{-2}$) | 12 (7.2~17) | 3.5 (1.6~8.4) | 3.1 (1.3~7.0 |

3 ________________________

3   Table.2 Comparisons of radiative forcing due to LAPs in snow (this study) with

4   observed and model-simulated values from previous studies

| Study | Region | Time period | Method | Radiative forcing (W m$^{-2}$) |
|---|---|---|---|---|
| Miller et al. (2016) | San Juan Mountains | May, 2010 | Remote sensing | ~37-100 |
| Sterle et al. (2013) | eastern Sierra Nevada | Feb to May, 2009 | In-situ measurements | ~2.5-40 |
| Miller et al. (2016) | San Juan Mountains | May, 2010 | In-situ measurements | 35-86 |
| Dang et al. (2017) | Northern China | Jan and Feb, 2010 and 2012 | In-situ measurements | 7–18 |

| | | | | |
|---|---|---|---|---|
| | North America | Jan-Mar, 2013-2014 | In-situ measurements | 0.6–1.9 |
| | The Arctic | Spring, 2005-2009 | In-situ measurements | 0.1–0.8 |
| Hansen and Nazarenko (2004) | North Hemisphere | | Model simulations | 0.3 |
| Qian et al. (2009) | western United States | Mar | Model simulations | ~3-7 |
| Bond et al. (2013) | Global | industrial era | Model simulations | 0.13 |
| Flanner et al. (2007) | Global | Annual 1998 (strong) | Model simulations | 0.054 |
| | | Annual 2001(weak) | | 0.049 |
| Qian et al. (2014) | Northeastern China | Apr | Model simulations | 5-10 |
| | North America | Apr | Model simulations | 2-7 |
| | The Arctic | Apr | Model simulations | <0.3 |
| Zhao et al. (2014) | Northeastern China | Jan and Feb, 2010 | Model simulations | 10 |
| Oaida et al. (2015) | western US | Spring, 2009-2013 | Model simulations | 16 |
| Qi et al. (2017) | The Arctic | Apr, 2008 | Model simulations | 0.024-0.39 |
| This study | Northeastern China | Dec-May, 2003-2018 | Remote sensing | 12 |
| | NA | | | 3.1 |
| | Canadian Arctic | | | 2.6 |
| | Russian Arctic | | | 3.3 |
| | Greenland | | | 1.3 |
| | EUA | | | 3.5 |

---

## Referee Report (RR1)

**General comments**

I thank the authors for their substantial efforts in revising the manuscript. Most of my original comments have been addressed satisfactorily. However, a few minor points should still be addressed, along with some technical/language corrections.

**Specific comments**

1. p. 2, line 13: I think it should be clarified what the stated value (2.9 W m$^{-2}$) actually represents. E.g., "This value represents land areas with complete or near-complete snow cover, with little or no vegetation above the snow."

2. p. 13, line 16: For Eq. (2) to be correct, $E_{\mathrm{dif}}^{\mathrm{clear}}$ should be the diffuse spectral irradiance on a *horizontal surface* and $E_{\mathrm{dir}}^{\mathrm{clear}}$ the direct spectral irradiance on a *surface perpendicular to the sun*. Please state this in the text, and importantly, check that this is indeed what SBDART provides.

3. p. 25, line 16: "significant altitude-dependent" trend? This requires a bit more explanation, e.g. are the values increasing or decreasing with altitude in the Russian Arctic?

4. p. 28, lines 4–18: It seems to me that the first factor listed here might be the best candidate for explaining why the ratio $\Delta\alpha_{\mathrm{MODIS,daily}}^{\mathrm{LAP}}$ / $\Delta\alpha_{\mathrm{in\text{-}situ,daily}}^{\mathrm{LAP}}$ tends to be larger than 1, especially for relatively clean snow (the last three factors also cause errors, but it is not obvious whether they usually give rise to an overestimate or underestimate). Any vegetation in the MODIS scene likely reduces the derived albedo, and this probably also applies to the effect of snow surface roughness (Manninen et al.: Effect of small-scale snow surface roughness on snow albedo and reflectance, The Cryosphere Discussions, https://doi.org/10.5194/tc-2020-154, in review, 2020.). To my understanding this cannot be accounted for in the pure snow albedo calculation with SNICAR, which might give rise to a positive albedo bias compared to that derived from MODIS — yielding therefore an overestimate of the albedo reduction attributed to LAPs?

5. Figure 1: In each panel, one parameter is varied while three are kept constant. What were the constant (i.e. default) values assumed in this figure?

6. Table S1: In addition to MAE and RMSE, it would be useful to give the correlation coefficient between the corrected MODIS retrievals and the measurement-based albedo reductions.

**Technical and language corrections**

1. p. 3, line 8: Replace "radiances" with "radiation". Also on p. 22, line 2.

2. p. 9, line 18: "which briefly". Something missing here?

3. p. 11, line 6: Replace "is performed" with "are assumed".

4. p. 15, line 4: Replace "competent" with "component".

5. p. 20, lines 15-17: Reformulation suggested, to improve clarity: "... and the results mainly represent winter for midlatitudes (because spring is mostly snow-free) and spring for the Arctic (because albedos cannot be derived during polar night)".

6. p. 21, line 2: "where is considerably higher". Something is missing here. Should it be "where the emissions are considerably higher"?

7. p. 22, line 4: Remove "radiances", or replace it with "radiative", since "radiances flux" is not correct. "Radiance" refers to the intensity of radiation coming from a certain direction, and "radiative flux" (aka. "irradiance") refers to the power radiated through a certain area, i.e., radiances integrated over a half-sphere.

8. p. 27, line 16 (and Fig. 6 and Fig. S8): The terms "negative uncertainty" and "positive uncertainty" are not commonly used. Do you mean "the lower bound and the upper bound of the uncertainty range"?

9. p. 27, lines 17-18: replace "by higher uncertainties" with "contributing more to the uncertainty".

10. p. 28, lines 7-11: This is not expressed very clearly. What about: "MODIS has variably spaced and discrete spectral bands and thus cannot provide a continuous spectral measurement of reflectance. This results in a non-negligible uncertainty in retrieving the radiative forcing by LAPs in snow."

11. p. 28, line 13: Should this be "a sample site located somewhere within the pixel"? (In-situ measurements are not necessarily taken at the midpoint of MODIS

pixels).

12. p. 28, line 14: replace "true" with "representative".

13. p. 30, line 2: replace "radiances" with "radiative fluxes".

14. p. 31, line 13, and p. 35, lines 8-9: replace "Earth system modeling" with "CESM2". (The performance of other Earth System Models might well differ from CESM2).

15. Fig. 7: Thank you for including this figure! To improve its readability, please consider using a colour scale with other colours than just red and white.

16. Fig. 9 (upper panel): The geographic factor (G) seems not to appear at all in the colour bars. Is this an error, or is the contribution too small to be seen?

17. Several of the figures in the Supplementary material (specifically, Figs. S1, S3, S4 and S6) would benefit from making the figure panels larger. Currently, a magnifying glass is required for reading the axis labels!

---

## Editor Decision (ED1)

[revised manuscript text omitted]

$$\Delta\alpha^{LAPs,fit}_{MODIS} = a \cdot \Delta\alpha^{LAPs}_{MODIS,corrected}(I_{LAPs}) + b\Delta\alpha^{LAPs}_{MODIS,corrected}(R_{eff}) + c \cdot$$
$$\Delta\alpha^{LAPs}_{MODIS,corrected}(SD) + d \cdot \Delta\alpha^{LAPs}_{MODIS,corrected}(G) \tag{18}$$

where $\Delta\alpha^{LAPs,fit}_{MODIS}$ is the fitted snow albedo reduction and a, b, c, and d denote the regression coefficients. Figure S3a illustrates how $\Delta\alpha^{LAPs,fit}_{MODIS}$ can explain 99% of the variance in $\Delta\alpha^{LAPs}_{MODIS,corrected}$. Therefore, the attribution of spatial variance in $\Delta\alpha^{LAPs}_{MODIS,corrected}$ can be replaced with $\Delta\alpha^{LAPs,fit}_{MODIS}$, enabling Eq. (18) to be written as follows:

$$\Delta\alpha^{LAPs,fit}_{MODIS} - \overline{\Delta\alpha^{LAPs,fit}_{MODIS}} = a \cdot \left(\Delta\alpha^{LAPs}_{MODIS,corrected}(
[revised manuscript text omitted]

[Figure]

**Figure S3.** Spatial distribution of (a) in-situ measurements of $BC_{equiv}$ and (b) the in situ snow albedo reduction and (c) radiative forcing. The snow albedo reduction and radiative forcing were calculated by SNICAR using measured $BC_{equiv}$.

[Figure]

**Figure S4.** Ratio of $\Delta\alpha_{MODIS,ins}$ to $\Delta\alpha_{in-situ,ins}$. Panels (a)–(f) represent the snow samples collected in Greenland, Russian Arctic, Canadian Arctic, NA, NWC, and NEC, respectively.

[Figure]

**Figure S5.** Comparisons of (a) $\Delta\alpha_{MODIS}$ and fitted albedo reduction ($\Delta\alpha_{Regression}$), and (b) $RF_{MODIS}$ and fitted radiative forcing ($RF_{Regression}$). Different colors represent different regions.

[Figure]

**Figure S6.** Spatial distributions of the lower bound of the uncertainty range due to (a) atmospheric correction, (b) snow cover fraction calculation and (c) snow grain size retrieval, respectively.

(d)-(f) Same as (a)-(b), but for the upper bound of the uncertainty range.

[Figure]

**Figure S7.** Statistics of daily radiative forcing, based on the CESM2 soot content of snow in December–May during the period 2003–2014. The boxes denote the 25th and 75th quantiles, and the horizontal lines represent the 50th quantiles (medians), the averages are shown as red dots; the whiskers denote the 5th and 95th quantiles.

Table S1. The mean absolute error (MAE) and root mean square error (RMSE) of $\Delta\alpha_{MODIS,corrected}^{LAPs}$ relative to

$\Delta\alpha_{situ,daily}^{LAPs}$

|  | Northeastern China | Northwestern China | NA | Canadian Arctic | Greenland | Russian Arctic |
|---|---|---|---|---|---|---|
| MAE | 0.064 | 0.016 | 0.014 | 0.0038 | 0.0014 | 0.011 |
| RMSE | 0.088 | 0.020 | 0.024 | 0.0075 | 0.0016 | 0.016 |
| Correlation coefficient | 0.13 | -0.22 | 0.25 | 0.53 | 0.37 | 0.27 |

**References**

Wang, R., Tao, S., Shen, H., Huang, Y., Chen, H., Balkanski, Y., Boucher, O., Ciais, P., Shen, G., Li, W., Zhang, Y., Chen, Y., Lin, N., Su, S., Li, B., Liu, J., and Liu, W.: Trend in global black carbon emissions from 1960 to 2007, Environ Sci Technol, 48, 6780-6787, 10.1021/es5021422, 2014.

---

## Author Response (AR2)

**Editor: Nikos Hatzianastassiou**

Thank the Editor very much for handling the manuscript. We take into account all the comments from referees and make revisions. Please check the response to the referees and the revised manuscript.

**Anonymous Referee #1**

We are very grateful for the referee's critical comments. The followings are our point-by-point responses to the comments. Our responses start with "R:".

**General comments**

I thank the authors for their substantial efforts in revising the manuscript. Most of my original comments have been addressed satisfactorily. However, a few minor points should still be addressed, along with some technical/language corrections.

R: Thank you very much for the positive comments, which will encourage us to do more in-depth research in the future. Moreover, the referee's comments and suggestions are quite significant that can help us to improve the paper quality substantially. We have addressed all of the comments carefully according to the suggestions. All of the detailed responses can be seen as follow.

**Specific comments**

1.  p. 2, line 13: I think it should be clarified what the stated value (2.9 W m$^{-2}$) actually represents. E.g., "This value represents land areas with complete or near-complete snow cover, with little or no vegetation above the snow."

    R: We have added the detailed description about what "2.9 W m$^{-2}$" actually represents as "…with daily radiative forcing ($RF_{MODIS,daily}^{LAPs}$) values of ~2.9 W m$^{-2}$, over land areas with complete or near-complete snow cover, with little or no vegetation above the snow in Northern Hemisphere." in p. 2, line 13-14 as suggestion.

2.  p. 13, line 16: For Eq. (2) to be correct, $E_{dif}^{clear}$ should be the diffuse spectral irradiance on a *horizontal surface* and $E_{dir}^{clear}$ the direct spectral irradiance on a *surface perpendicular to the sun*. Please state this in the text, and importantly, check that this is indeed what SBDART provides.

R: We have added the statement about $E_{dif}^{clear}$ and $E_{dir}^{clear}$ as "…$E_{dif}^{clear}(\lambda; \varphi)$ denote the diffuse spectral irradiance on a horizontal surface and $E_{dir}^{clear}(\lambda; \varphi)$ denote the direct spectral irradiance on a surface perpendicular to the sun…" in p. 13, line 17-19. And we determine that the direct and diffuse spectral irradiance are indeed provided by SBDART just like Painter et al. (2012).

3. p. 25, line 16: "significant altitude-dependent" trend? This requires a bit more explanation, e.g. are the values increasing or decreasing with altitude in the Russian Arctic?

R: Revised as suggestion in p. 25, line 16-17: "In Russian Arctic, $\Delta\alpha_{MODIS,corrected}^{LAPs}$ and $RF_{MODIS,daily}^{LAPs}$ values increase with altitude of ~0.012-0.048 and ~1.0-7.3 W m$^{-2}$."

4. p. 28, lines 4–18: It seems to me that the first factor listed here might be the best candidate for explaining why the ratio $\Delta\alpha_{MODIS,daily}^{LAPs}/\Delta\alpha_{in-situ,daily}^{LAPs}$ to be larger than 1, especially for relatively clean snow (the last three factors also cause errors, but it is not obvious whether they usually give rise to an overestimate or underestimate). Any vegetation in the MODIS scene likely reduces the derived albedo, and this probably also applies to the effect of snow surface roughness (Manninen et al.: Effect of small-scale snow surface roughness on snow albedo and reflectance, The Cryosphere Discussions, https://doi.org/10.5194/tc- 2020-154, in review, 2020.). To my understanding this cannot be accounted for in the pure snow albedo calculation with SNICAR, which might give rise to a positive albedo bias compared to that derived from MODIS — yielding therefore an overestimate of the albedo reduction attributed to LAPs?

R: As the referee said, the effect of snow surface roughness and vegetation, which were without regarding in SNICAR, probably reduce the derived albedo from MODIS and therefore result in overestimate of the albedo reduction attributed to LAPs (the ratio $\Delta\alpha_{MODIS,daily}^{LAPs}/\Delta\alpha_{in-situ,daily}^{LAPs}$ to be larger than 1).

5.  Figure 1: In each panel, one parameter is varied while three are kept constant. What were the constant (i.e. default) values assumed in this figure?

R: We have added the assumed constant variabilities values in each panel in Figure 1:

[Figure]

Figure 1. Variations in spectral snow albedo due to (a) LAP content (ng g$^{-1}$), (b) snow depth (m), (c) snow grain size (μm), and (d) solar zenith angle (deg.) while other three parameters are kept constant.

6. Table S1: In addition to MAE and RMSE, it would be useful to give the correlation coefficient between the corrected MODIS retrievals and the measurement-based albedo reductions.

R: We have added the correlation coefficient in Table S1:

| | Northeastern China | Northwestern China | NA | Canadian Arctic | Greenland | Russian Arctic |
|---|---|---|---|---|---|---|
| MAE | 0.064 | 0.016 | 0.014 | 0.0038 | 0.0014 | 0.011 |
| RMSE | 0.088 | 0.020 | 0.024 | 0.0075 | 0.0016 | 0.016 |
| Correlation coefficient | 0.13 | -0.22 | 0.25 | 0.53 | 0.37 | 0.27 |

**Technical and language corrections**

1. p. 3, line 8: Replace "radiances" with "radiation". Also on p. 22, line 2.
   R: Revised as suggestion.

2. p. 9, line 18: "which briefly". Something missing here?
   R: Sorry for the grammar mistake and we have revised as "The equivalent BC has been defined by Doherty et al. (2010) which briefly as the amount of BC in the snow accounted for the wavelength-integrated total light absorption in the wavelengths of 300-750 nm by all particulate constituents." in p. 9, line 18-21.

3. p. 11, line 6: Replace "is performed" with "are assumed".
   R: Revised as suggestion.

4. p. 15, line 4: Replace "competent" with "component".
   R: Revised as suggestion.

5. p. 20, lines 15-17: Reformulation suggested, to improve clarity: "... and the results mainly represent winter for midlatitudes (because spring is mostly snow-free) and spring for the Arctic (because albedos cannot be derived during polar night)".
   R: Revised as suggestion.

6. p. 21, line 2: "where is considerably higher". Something is missing here. Should it be "where the emissions are considerably higher"?

R: Revised as suggestion.

7. p. 22, line 4: Remove "radiances", or replace it with "radiative", since "radiances flux" is not correct. "Radiance" refers to the intensity of radiation coming from a certain direction, and "radiative flux" (aka. "irradiance") refers to the power radiated through a certain area, i.e., radiances integrated over a half-sphere.

   R: Thank you for explaining and distinguishing the concept about "radiances", "radiative", "radiances flux" and "irradiance". We have rechecked the errors and revised throughout the manuscript.

8. p. 27, line 16 (and Fig. 6 and Fig. S8): The terms "negative uncertainty" and "positive uncertainty" are not commonly used. Do you mean "the lower bound and the upper bound of the uncertainty range"?

   R: Sorry for the non-standard terminology and revised as suggestion.

9. p. 27, lines 17-18: replace "by higher uncertainties" with "contributing more to the uncertainty".

   R: Revised as suggestion.

10. p. 28, lines 7-11: This is not expressed very clearly. What about: "MODIS has variably spaced and discrete spectral bands and thus cannot provide a continuous spectral measurement of reflectance. This results in a non-negligible uncertainty in retrieving the radiative forcing by LAPs in snow."

    R: Revised as suggestion.

11. p. 28, line 13: Should this be "a sample site located somewhere within the pixel"? (In-situ measurements are not necessarily taken at the midpoint of MODIS pixels).

    R: Revised as suggestion.

12. p. 28, line 14: replace "true" with "representative".

    R: Revised as suggestion.

13. p. 30, line 2: replace "radiances" with "radiative fluxes".

    R: Revised as suggestion.

14. p. 31, line 13, and p. 35, lines 8-9: replace "Earth system modeling" with "CESM2". (The performance of other Earth System Models might well differ from CESM2).

R: Revised as suggestion and we would like to add more modeling simulations to compare with our retrievals if these datasets were available in CMIP6 in the future.

15. Fig. 7: Thank you for including this figure! To improve its readability, please consider using a colour scale with other colours than just red and white.

R: Revised as suggestion:

[Figure]

Figure 7. The spatial distribution of the ratio of retrieved radiative forcing using semi-infinite snow to radiative forcing using ERA-Interim snow depth.

16. Fig. 9 (upper panel): The geographic factor (G) seems not to appear at all in the colour bars. Is this an error, or is the contribution too small to be seen?

R: Actually, the geographic factor ($G$) makes too small contribution ($< 1\%$) to snow albedo reduction, both on regional and global scales.

17. Several of the figures in the Supplementary material (specifically, Figs. S1, S3, S4 and S6) would benefit from making the figure panels larger. Currently, a magnifying glass is required for reading the axis labels!

R: Revised as suggestion:

[Figure]

**Figure S1.** (a) Average December-May incident direct solar spectra for latitudes 35°–85°, derived from the SBDART model during clear-sky conditions. (b) Same as (a), but for diffuse solar irradiance. (c) Same as (a), but for cloudy-sky condition.

[Figure]

**Figure S3.** Spatial distribution of (a) in-situ measurements of $BC_{equiv}$ and (b) the in-situ snow albedo reduction and (c) radiative forcing. The snow albedo reduction and radiative forcing were calculated by SNICAR using measured $BC_{equiv}$.

[Figure]

**Figure S4.** Ratio of $\Delta\alpha_{MODIS,ins}$ to $\Delta\alpha_{in-situ,ins}$. Panels (a)–(f) represent the snow samples collected in Greenland, Russian Arctic, Canadian Arctic, NA, NWC, and NEC, respectively.

[Figure]

**Figure S6.** Spatial distributions of the lower bound of the uncertainty range due to (a) atmospheric correction, (b) snow cover fraction calculation and (c) snow grain size retrieval, respectively. (d)-(f) Same as (a)-(b), but for the upper bound of the uncertainty range.

**Edward Bair (Referee #3)**

nbair@eri.ucsb.edu

We are very grateful for the referee's constructive comments and suggestions throughout the peer-review process. Moreover, the referee's comments are quite significant that can help us to improve the paper quality substantially, especially in the uncertainty discussions and data availability. We have already rechecked the manuscript and made grammatical corrections and technical corrections.

[revised manuscript text omitted]

---

## Author Response (AR3)

**Editor: Nikos Hatzianastassiou**

Thank the Editor very much for handling the manuscript. We have taken into account all the comments and suggestions from the Editor's review (manuscript plus supplement) and made technical revisions. Please check the response, the revised manuscript and the revised supplement. The followings are our point-by-point responses to the comments. Our responses start with "R:".

**Technical and language corrections**

1. p. 2, line 12: Replace "condition" with "conditions".

    R: Revised as suggestion.

2. p. 2, line 13: Replace "cover, with" with "cover and with".

    R: Revised as suggestion.

3. p. 25, line 17: Replace "of" with "by".

    R: Revised as suggestion.

4. p. 28, line 5: Replace "It worth" with "It is worth".

    R: Revised as suggestion.

5. p. 28, line 8: Replace comma with full stop.

    R: Revised as suggestion.

6. p. 28, line 8: Replace "which were without regarding" with "This effect is not accounted for".

    R: Revised as suggestion.

7. p. 28, line 9: Replace "reduce" with "reduces".

    R: Revised as suggestion.

8. p. 28, line 9: Replace "result" with "results".

    R: Revised as suggestion.

9. p. 28, line 10: Replace "overestimate" with "an overestimation".

    R: Revised as suggestion.

10. Supplement, Table S1: Also add correlation coefficient in Figure caption.

    R: Revised as suggestion in both manuscript and supplement.

[revised manuscript text omitted]
^{LAPs}_{MODIS,corrected}(I_{LAPs})}) + b \cdot \left(\Delta\alpha^{LAPs}_{MODIS,corrected}(R_{eff}) - \right.$$

$$\overline{\Delta\alpha^{LAPs}_{MODIS,corrected}(R_{eff})}) + c \cdot \left(\Delta\alpha^{LAPs}_{MODIS,corrected}(SD) - \right.$$

$$\overline{\Delta\alpha^{LAPs}_{MODIS,corrected}(SD)}) + d \cdot \left(\Delta\alpha^{LAPs}_{MODIS,corrected}(G) - \overline{\Delta\alpha^{LAPs}_{MODIS,corrected}(G)}\right) \quad (19)$$

where $\Delta\alpha^{LAPs,fit}_{MODIS} - \overline{\Delta\alpha^{LAPs,fit}_{MODIS}}$ is the snow albedo reduction anomaly

$(\Delta\alpha^{LAPs,fit}_{MODIS,anomaly})$. Then, Eq. (19) can be written as

$$\Delta\alpha^{LAPs,fit}_{MODIS,anomaly} = a \cdot \Delta\alpha^{LAPs}_{MODIS,corrected,anomaly}(I_{LAPs}) + b \cdot$$

$$\Delta\alpha^{LAPs}_{MODIS,corrected,anomaly}(R_{eff}) + c \cdot \Delta\alpha^{LAPs}_{MODIS,corrected,anomaly}(SD) + d \cdot$$

[revised manuscript text omitted]

(d)-(f) Same as (a)-(b), but for the upper bound of the uncertainty range.

[Figure]

**Figure S7.** Statistics of daily radiative forcing, based on the CESM2 soot content of snow in December–May during the period 2003–2014. The boxes denote the 25th and 75th quantiles, and the horizontal lines represent the 50th quantiles (medians), the averages are shown as red dots; the whiskers denote the 5th and 95th quantiles.

Table S1. The mean absolute error (MAE),  root mean square error (RMSE) and correlation coefficient of

$\Delta\alpha_{MODIS,corrected}^{LAPs}$ relative to $\Delta\alpha_{in-situ,daily}^{LAPs}$

|  | Northeastern China | Northwestern China | NA | Canadian Arctic | Greenland | Russian Arctic |
|---|---|---|---|---|---|---|
| MAE | 0.064 | 0.016 | 0.014 | 0.0038 | 0.0014 | 0.011 |
| RMSE | 0.088 | 0.020 | 0.024 | 0.0075 | 0.0016 | 0.016 |
| Correlation coefficient | 0.13 | -0.22 | 0.25 | 0.53 | 0.37 | 0.27 |

**References**

Wang, R., Tao, S., Shen, H., Huang, Y., Chen, H., Balkanski, Y., Boucher, O., Ciais, P., Shen, G., Li, W., Zhang, Y., Chen, Y., Lin, N., Su, S., Li, B., Liu, J., and Liu, W.: Trend in global black carbon emissions from 1960 to 2007, Environ Sci Technol, 48, 6780-6787, 10.1021/es5021422, 2014.